# AnyPos: Automated Task-Agnostic Actions for Bimanual Manipulation

## Abstract

Learning generalizable manipulation policies hinges on data, yet robot manipulation data is scarce and often entangled with specific embodiments, making both cross-task and cross-platform transfer difficult. We tackle this challenge with **task-agnostic embodiment modeling**, which learns embodiment dynamics directly from *task-agnostic action* data and decouples them from high-level policy learning. By focusing on exploring all feasible actions of the embodiment to capture what is physically feasible and consistent, task-agnostic data takes the form of independent image-action pairs with the potential to cover the entire embodiment workspace, unlike task-specific data, which is sequential and tied to concrete tasks. This data-driven perspective bypasses the limitations of traditional dynamics-based modeling and enables scalable reuse of action data across different tasks. Building on this principle, we introduce **AnyPos**, a unified pipeline that integrates large-scale automated task-agnostic exploration with robust embodiment modeling through inverse dynamics learning. AnyPos generates diverse yet safe trajectories at scale, then learns embodiment representations by *decoupling arm and end-effector motions* and employing a *direction-aware decoder* to stabilize predictions under distribution shift, which can be seamlessly coupled with diverse high-level policy models. In comparison to the standard baseline, AnyPos achieves a 51% improvement in test accuracy. On manipulation tasks such as operating a microwave, toasting bread, folding clothes, watering plants, and scrubbing plates, AnyPos raises success rates by 30–40% over strong baselines. These results highlight data-driven embodiment modeling as a practical route to overcoming data scarcity and achieving generalization across tasks and platforms in visuomotor control.

## 1    Introduction

Building embodied agents that can perceive, reason, and act in complex physical environments remains a central goal of robotics and AI. Vision–language–action (VLA) models such as RT-X O'Neill et al. (2024), Octo Ghosh et al. (2024), RDT Liu et al. (2024), and OpenVLA Kim et al. (2024) advance this goal by learning task-conditioned visuomotor policies from paired demonstrations, achieving impressive results in tasks like pick-and-place or instruction following Kim et al. (2024); Liu et al. (2024). Yet, their ability to generalize remains fundamentally constrained by data. Robotic datasets are expensive to curate, often tightly coupled to specific hardware, and predominantly *task-specific*: they concentrate on narrow goal distributions (e.g., stacking blocks, opening doors) within fixed embodiments. Such data under-covers the state–action space, limits behavioral diversity, and fails to transfer across morphologies—an issue widely documented in benchmarks such as ManiSkill2 Gu et al. (2023), RT-X O'Neill et al. (2024), and RoboVerse Geng et al. (2025), and underscored by large-scale efforts like Bridge Data Ebert et al. (2022).

In this work, we take a complementary route through *task-agnostic embodiment modeling*. Rather than supervising policies with goal labels, we exploit trajectories that capture the task-invariant structure of body–world interaction—kinematics, reachability, and contact dynamics. This reframes the learning problem from "what actions should be taken to accomplish a labeled goal " to "what actions are physically feasible and consistent." By shifting focus to feasibility through the leverage of diverse embodiment-specific data, embodiment modeling supplies reusable priors that expand coverage of the state–action space, reduce dependence on narrow goal annotations, and transfer across tasks, embodiments, and viewpoints.

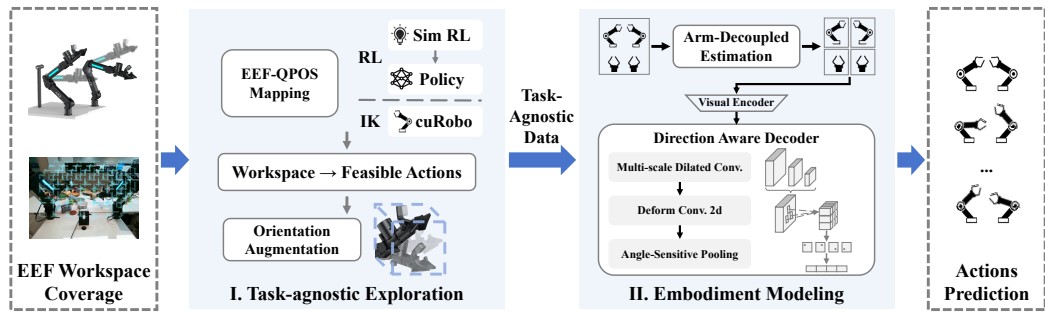

Figure 1: **AnyPos illustration.** We obtain a task-agnostic dataset covering the entire feasible cubic workspace of robotic arms for embodiment modeling. **Input to AnyPos**: Images containing the robotic arms. **Output of AnyPos**: The action/joint position values inferred from the image.

Crucially, embodiment data and task-specific data are not substitutes but complements. Unlabeled embodiment-specific trajectories capture *what is feasible*, supporting dynamics and inverse mappings (e.g., $p(s_{t+1} \mid s_t, a_t)$, $p(a_t \mid s_t, s_{t+1})$), while goal-conditioned demonstrations capture *what is desired* (e.g., $p(a_t \mid s_t, g)$ or $p(a_t \mid s_t, \ell)$). Decoupling feasibility from desirability yields two benefits: (1) few-shot adaptation, where a lightweight goal module can be trained atop a stable embodiment backbone, and (2) rollout stability, as long-horizon predictions are gated by feasibility checks learned from task-agnostic data. In this framing, labels are reserved for *which/why*, while embodiment modeling supplies the *how*, reducing data costs and enabling scalable generalization across tasks and platforms.

Following the above motivation, we instantiate *task-agnostic embodiment modeling* with **AnyPos**, a unified framework that learns reusable embodiment priors transferable across tasks. AnyPos emphasizes feasibility—"what actions are physically consistent and executable"—rather than direct goal achievement, and is instantiated through a two-step pipeline complemented by an extensible design for coupling with higher-level policies, as demonstrated in Fig. 1.

*First*, we automate task-agnostic exploration to collect diverse, safety-aware, and feasible trajectories without relying on goal labels or human teleoperation. To fully cover the manipulator's 3D workspace, we employ a three-stage approach. Initially, we construct a mapping from end-effector positions to feasible joint positions using either reinforcement learning or inverse kinematics. Next, the embodiment-specific mapping guides a uniform exploration of the workspace. Finally, we further enrich the collected data through orientation augmentation for the wrist joints. This procedure yields large-scale, physically grounded ⟨image, action⟩ pairs that expand the state–action space beyond goal-specific demonstrations. *Second*, we learn inverse dynamics from these unlabeled rollouts using lightweight inductive biases that stabilize training on noisy, task-agnostic data. To be more specific, the model takes in an image and predicts the actions of the robot depicted in it. Concretely, we *decouple* the robot into separate components (e.g., each arm and end-effector) to suppress irrelevant joints and disentangle cross-arm effects, and we employ a *direction-aware decoder* that aligns visual features with plausible motion directions, improving robustness under distribution shift. Together, AnyPos replaces supervision about "what actions should be taken to achieve a goal" with supervision about "what is physically feasible and consistent." The resulting embodiment backbone is modular: it can be seamlessly coupled with various high-level policy models—such as goal-conditioned or video-conditioned models—enabling few-shot adaptation and stable rollout without redesigning the low-level dynamics.

**Results.** Our experiments demonstrate that this perspective translates into both stronger embodiment modeling and tangible task-level gains. AnyPos achieves significantly higher accuracy in action prediction on challenging test sets with unseen skills and objects, surpassing standard baselines by over 51%. When deployed to real robots, the learned embodiment backbone further improves manipulation success rates by more than 30% compared to models trained on human-collected datasets. Moreover, AnyPos is modular: when coupled with complementary models such as diffusion-based video generation models, it extends naturally to diverse tasks including basket lifting, clicking, and pick-and-place with unseen objects. These results highlight the advantage of framing embodiment modeling as learning *what is physically feasible and consistent*, and establish AnyPos as a scalable foundation for generalizable visuomotor control.

## 2 RELATED WORK

**Embodied Data Collection.** Data collection for embodied AI typically falls into three categories: simulation, real robots, and internet videos. Simulation-based approaches such as RoboTwin (Mu et al., 2024), ManiBox (Tan et al., 2024), and AgiBot DigitalWorld (Zhang et al., 2025) enable scalable collection at low cost, but face persistent Sim2Real gaps and limited physical fidelity on complex manipulation tasks. Real-world pipelines, including Diffusion Policy (Chi et al., 2023), Mobile Aloha (Fu et al., 2024), recent VLAs (Liu et al., 2024; O'Neill et al., 2024; Kim et al., 2024), and large-scale datasets (Khazatsky et al., 2024; Ebert et al., 2022; Wu et al., 2024; AgiBot-World-Contributors et al., 2025), demonstrate strong practical capabilities but remain expensive and constrained by task-specific action labels, which hinder generalization across embodiments. Internet videos, by contrast, offer abundant priors on physical interactions and motion patterns, and early work (Du et al., 2023; Hu et al., 2024; Cheang et al., 2024; Zhou et al., 2024) shows promise in leveraging them. Yet connecting raw video to high-precision action generation is still an open challenge.

**Embodied Policies and VLAs.** Recent embodied manipulation policies such as ACT (Fu et al., 2024) and Diffusion Policy (Chi et al., 2023; Ze et al., 2024; Ren et al., 2024) have achieved success in real-world tasks, learning direct mappings from visual input to action trajectories. However, these policies are largely single-task and lack explicit language grounding or multi-task scalability. To address this, vision-language-action (VLA) models (Liu et al., 2024; Zitkovich et al., 2023; Brohan et al., 2022; Ghosh et al., 2024; Kim et al., 2024; Liu et al., 2025; Ding et al., 2025; Li et al., 2024a; O'Neill et al., 2024; Pertsch et al., 2025; Black et al., 2024) introduce natural language as a task-conditioning signal, enabling broader instruction following and multi-task generalization. Despite their promise, VLAs depend on large-scale, task-conditioned action datasets for each embodiment. Current datasets remain relatively small and embodiment-specific, leaving persistent gaps in generalization and limiting robustness under morphology shifts (O'Neill et al., 2024).

**Embodiment Modeling for Manipulation.** A key gap is *embodiment modeling*—learning morphology-specific feasibility priors that transcend tasks. Cross-embodiment datasets and generalist policies (Open-X Embodiment, RT-X, Octo) improve transfer but still entangle task semantics with embodiment constraints (O'Neill et al., 2024; Zitkovich et al., 2023; Ghosh et al., 2024). World-model and generative lines (UniSim, RoboDreamer) and planners built on predicted futures (UniPi, Gen2Act, VPP, Seer/PIDM) broaden flexibility but face inconsistencies across action spaces and reliance on task-labeled actions (Yang et al., 2024; Zhou et al., 2024; Du et al., 2023; Bharadhwaj et al., 2024; Hu et al., 2024; Tian et al., 2024). Generalist agents and curated multi-env datasets (RoboCat, BridgeData V2) report cross-robot adaptation, yet require demonstrations and platform tuning (Bousmalis et al., 2024; Walke et al., 2023). These limitations motivate *task-agnostic embodiment modeling*: learning a reusable inverse-dynamics prior from unlabeled exploration that decouples feasibility from semantics and supports precise, stable control across morphologies.

## 3 METHOD

### 3.1 TASK-AGNOSTIC EMBODIMENT MODELING

We consider language-conditioned robotic manipulation with observation $x \in \mathcal{X}$, instruction $\ell \in \mathcal{L}$, and action $a \in \mathcal{A}$. Here, $\mathcal{X}, \mathcal{A} \subseteq \mathbb{R}^d$ and $\mathcal{L}$ denote the observation, action, and language command spaces, respectively, where $d$ denotes the dimensionality of the action. Here, $\mathcal{X}$ and $\mathcal{A} \subseteq \mathbb{R}^d$ denote the observation space and the action space, respectively, where $d$ denotes the dimensionality of the action. For example, for a 6-DoF dual-arm manipulator with two grippers, $\mathcal{A} \subseteq \mathbb{R}^{14}$.

The agent learns a policy $\pi$ that takes $x$ and $\ell$ and rolls out $a$ to complete the task. Standard VLA models learn temporally extended policies $p_\theta(a_{T+1:T+k} \mid x_{T-H+1:T}, \ell)$, [1] where $\theta$ are model parameters, $T$ is the current timestep, $k$ is the action chunk size (Zhao et al., 2023), and $H$ is the history window, which is typically set to 1. Given an expert dataset $D_{\text{expert}}$, the training objective maximizes

$$\max_\theta \ \mathbb{E}_{a_{T+1:T+k}, x_T, \ell \sim D_{\text{expert}}} \ p_\theta(a_{T+1:T+k} \mid x_T, \ell). \tag{1}$$

---

[1] For clarity, we denote the model's action at timestep $i-1$ as $a_i$, which corresponds to the joint position at timestep $i$.

However, due to the high-dimensional nature of $(\mathcal{L}, \mathcal{A}^k)$, such direct modeling is data-hungry and brittle.

**Task-agnostic factorization.** Following a feasibility-first view, we factor action prediction by integrating over all possible future:

$$p(\boldsymbol{a}_{T+1:T+k} \mid \boldsymbol{x}_T, \ell) = \int p(\boldsymbol{x}_{T+1:T+k} \mid \boldsymbol{x}_T, \ell)\, p(\boldsymbol{a}_{T+1:T+k} \mid \boldsymbol{x}_{T+1:T+k})\, d\boldsymbol{x}_{T+1:T+k} \quad (2)$$

$$= \mathbb{E}_{\boldsymbol{x}_{T+1:T+k} \sim p(\boldsymbol{x}_{T+1:T+k} \mid \boldsymbol{x}_T, \ell)} \left[ \prod_{i=T+1}^{T+k} p(\boldsymbol{a}_i \mid \boldsymbol{x}_{i-1}, \boldsymbol{x}_i) \right]. \quad (3)$$

For position-controlled robots, $\boldsymbol{a}_i$ depends solely on $\boldsymbol{x}_i$, so $p(\boldsymbol{a}_i \mid \boldsymbol{x}_{i-1}, \boldsymbol{x}_i)$ reduces to $p(\boldsymbol{a}_i \mid \boldsymbol{x}_i)$. Even if the action space includes joint velocities, conditioning on $\boldsymbol{x}_{i-1}$ suffices. This yields a decomposition into *task-specific predicted images* and *task-agnostic actions*:

$$\underbrace{p(\boldsymbol{a}_{T+1:T+k} \mid \boldsymbol{x}_T, \ell)}_{\text{task-specific actions}} = \mathbb{E}_{\boldsymbol{x}_{T+1:T+k} \sim p(\boldsymbol{x}_{T+1:T+k} \mid \boldsymbol{x}_T, \ell)} \left[ \prod_{i=T+1}^{T+k} \underbrace{p(\boldsymbol{a}_i \mid \boldsymbol{x}_{i-1}, \boldsymbol{x}_i)}_{\text{task-agnostic actions}} \right]. \quad (4)$$

**AnyPos: Modular Embodiment Modeling.** We introduce AnyPos, a framework for task-agnostic embodiment modeling that separates semantic intent from physical feasibility. At its core, an action prediction model $F_\delta$ is pre-trained on large-scale, unlabeled exploration data $D_{\text{agnostic}} = \{(\boldsymbol{x}_{i-1}, \boldsymbol{a}_i, \boldsymbol{x}_i)\}$. The model learns to map observation transitions $(\boldsymbol{x}_{i-1}, \boldsymbol{x}_i)$ or observation $\boldsymbol{x}_i$ into feasible actions $\boldsymbol{a}_i$ by minimizing an action-space discrepancy:

$$\min_\delta \; \mathbb{E}_{(\boldsymbol{x}_i, \boldsymbol{a}_i) \sim \mathcal{D}_{\text{agnostic}}} \; d\big(\boldsymbol{a}_i, \mathcal{F}_\delta(\boldsymbol{x}_{i-1}, \boldsymbol{x}_i)\big), \quad (5)$$

where $d : \mathcal{A} \times \mathcal{A} \to \mathbb{R}^+$ is an action-space metric. Through this pre-training on a broad range of feasible actions, the model $F_\delta$ acquires a fundamental ability to generalize across the action space, producing smooth, physically valid behaviors (e.g., collision avoidance, stable motions) independent of downstream tasks—effectively serving as a form of embodiment modeling.

This universal feasibility prior can be seamlessly coupled with high-level policies (e.g., video generation models, VLAs, world models) that predict task-aligned future features, via co-training or model pipelines; $F_\delta$ then grounds these predictions into executable actions. By learning a "shared motor library" (i.e., prior knowledge of feasible action space) from large-scale, inexpensive, unlabeled action data, AnyPos reduces reliance on costly human demonstrations, and enables generalist policies to adapt to new skills and tasks with strong, zero-shot generalization.

## 3.2 AUTOMATED EXPLORATION FOR TASK-AGNOSTIC ACTION DATA COLLECTION

To instantiate the task-agnostic factor , we need large volumes of diverse yet *safe* trajectories collected *without* teleoperation or goal labels. Pure joint-space randomization underperforms in practice, yielding poor coverage and frequent self-collisions (Fig. 7). AnyPos reframes exploration as *feasible-action synthesis*: uniformly sample end-effector (EEF) targets in workspace and project each target to a collision-free joint configuration, thereby turning uniform task-space coverage into physically grounded actions. While this projection could be achieved using either IK or an RL policy, we adopt a task-agnostic RL policy to avoid the physically infeasible solutions that IK can sometimes produce. Notably, the RL policy is used only for projecting EEF targets to joint positions.

Let the reachable EEF workspace be a bounded volume $\mathcal{W} \subset \mathbb{R}^3$ and the action space be joint positions $\mathcal{A} \subset \mathbb{R}^d$. AnyPos learns $f_{\text{RL}} : \mathcal{W} \to \mathcal{A}$ that maps a target $\boldsymbol{w} \in \mathcal{W}$ to a feasible action. We adopt position control and simplify $p(\boldsymbol{a}_i \mid \boldsymbol{x}_{i-1}, \boldsymbol{x}_i)$ to $p(\boldsymbol{a}_i \mid \boldsymbol{x}_i)$; extensions to velocity/torque control are analogous. A policy $\pi_\theta(\boldsymbol{a} \mid \boldsymbol{w})$ is trained in simulation with PPO to minimize target error subject to safety:

$$r(\boldsymbol{a}; \boldsymbol{w}) = -\|\boldsymbol{w} - \boldsymbol{w}_{target}\|_2^2 \; - \; \gamma\, \phi_{\text{coll}}(\boldsymbol{a}) \; - \; \eta\, \phi_{\text{limit}}(\boldsymbol{a}),$$

where $x(\boldsymbol{a})$ is the forward-kinematics EEF position, $\phi_{\text{coll}}$ penalizes self/scene proximity, and $\phi_{\text{limit}}$ penalizes joint/velocity violations. At rollout, samples from $\mathcal{W}$ are projected to feasible actions by $f_{\text{RL}}$ and executed to log $(\boldsymbol{x}_i, \boldsymbol{a}_i, \boldsymbol{x}_{i+1})$.

The exploration process maintains a voxel grid over $\mathcal{W}$ and selects EEF targets using low-discrepancy sequences with inverse-visit reweighting, ensuring balanced coverage and a curriculum that expands gradually from a compact core to the full workspace. Each target is then projected into a constraint-compliant joint configuration via $f_{\mathrm{RL}}$, guaranteeing feasibility under kinematic and safety constraints. To enrich contact diversity, orientation-related joints are sampled from $\mathcal{A}_{\mathrm{wrist}}$ and appended to the RL output, yielding $\boldsymbol{a}_{\mathrm{aug}} = [\, f_{\mathrm{RL}}(\boldsymbol{w}) \,\|\, \boldsymbol{a}_{\mathrm{wrist}} \,]$. Execution is further protected by a real-time safety shield that enforces bounded-rate increments, distance margins, and actuator-current thresholds.

**Bimanual embodiments.** For dual-arm platforms, we introduce a minimal spatial prior via a random separating plane $\mathcal{B}$ that partitions $\mathcal{W}$ into $(\mathcal{W}_L, \mathcal{W}_R)$. Independently sample $\boldsymbol{w}_L \sim \mathcal{U}(\mathcal{W}_L)$ and $\boldsymbol{w}_R \sim \mathcal{U}(\mathcal{W}_R)$, map them to $(\boldsymbol{a}_L, \boldsymbol{a}_R)$ with $f_{\mathrm{RL}}$, and apply coupled collision checks; violations trigger resampling. This preserves breadth while preventing inter-arm interference.

AnyPos factorizes exploration into *workspace coverage* and *feasibility projection*. Uniform sampling in $\mathcal{W}$ guarantees broad behavioral support, while $f_{\mathrm{RL}}$ anchors each sample in physical constraints. Orientation enrichment expands contact modes without destabilizing reachability, and the bimanual prior injects just enough coordination to avoid collisions while keeping data task-agnostic. The result is dense, collision-aware $\langle \text{image}, \text{action} \rangle$ pairs that faithfully encode embodiment constraints.

**Embodiment-aware reuse.** AnyPos depends only on the robot URDF and kinematics, not on camera intrinsics/extrinsics or scene semantics. When sensors or viewpoints change, we simply replay workspace sampling and feasibility projection to regenerate trajectories consistent with the new setup, preserving embodiment constraints and enabling rapid data refresh across platforms.

Compared to naive joint-space sampling, AnyPos attains markedly better workspace coverage with substantially fewer collisions, and scales seamlessly from single- to dual-arm systems under the same policy and safety shield. The resulting task-agnostic dataset forms a strong prior for downstream policy learning, where semantics can be injected later through video or instruction alignment.

## 3.3 Embodiment Modeling and Applying Task Semantics

We train our model $\mathcal{F}_\delta$ on task-agnostic dataset $\mathcal{D}_{\mathrm{agnostic}}$ to learn a feasibility prior:

$$\min_{\delta} \ \mathbb{E}_{(\boldsymbol{x}_i, \boldsymbol{a}_i) \sim \mathcal{D}_{\mathrm{agnostic}}} \ d\big( \boldsymbol{a}_i, \ \mathcal{F}_\delta(\boldsymbol{x}_{i-1}, \boldsymbol{x}_i) \big), \qquad (6)$$

where $d(\cdot, \cdot)$ is a regression loss. When the entire arm configuration is visible and the platform uses position control, we adopt a deterministic mapping $\mathcal{F}_\delta : \mathcal{X} \to \mathcal{A}$; otherwise we condition on two frames, $\mathcal{F}_\delta : \mathcal{X}^2 \to \mathcal{A}$ with inputs $(\boldsymbol{x}_{i-1}, \boldsymbol{x}_i)$.

### 3.3.1 Training with Task-Agnostic Data

Let $\boldsymbol{x}$ denote multi-view observations (e.g., overhead and wrist cameras) and $\boldsymbol{a} = (a_1, \ldots, a_d)$ the joint configuration. For dual 6-DoF arms with grippers, $d = 14$. Direct monolithic regression is fragile due to doubled output dimensionality, combinatorial joint hypotheses, cross-arm visual interference, and the high precision (See Fig. 2) required for reliable replay. We therefore combine **arm-decoupled estimation** with a **Direction-Aware Decoder (DAD)**.

**Arm-decoupled estimation.** A heuristic segmentation $\Phi : \boldsymbol{x} \to (\boldsymbol{x}_L, \boldsymbol{x}_R)$ (initialized by pedestal/shoulder seeds with a split fallback under occlusion) isolates each arm; we then regress joints independently:

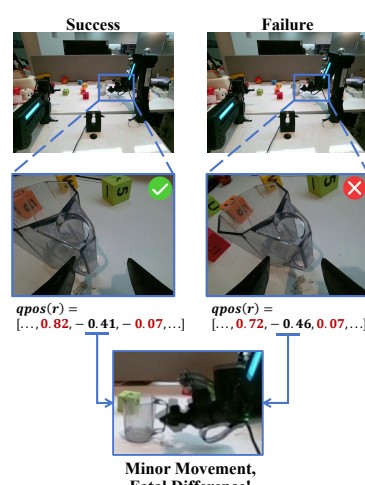

Figure 2: A visual example of the high precision requirements for robotic manipulation. A minor movement in just one dimension can lead to the failure of the entire operation. This level of precision presents a formidable challenge for action estimation.

$$\boldsymbol{x} \xrightarrow{\Phi} (\boldsymbol{x}_L, \boldsymbol{x}_R) \xrightarrow{f_L, f_R} \hat{\boldsymbol{a}} = \big[ f_L(\boldsymbol{x}_L) \,;\, f_R(\boldsymbol{x}_R) \big],$$

with grippers predicted by wrist-centric heads. Decoupling reduces cross-arm interference (see Appendix A.2)) and narrows the hypothesis space.

**Direction-Aware Decoder (DAD).** Using a DINOv2-with-registers encoder (DINOv2-Reg) for clean, spatially faithful features, DAD targets sub-0.06 joint error (on a 3.0-unit scale) via three components: (i) *Multi-scale dilated convs* $F_d = \sigma(\mathcal{C}_d(\boldsymbol{Y}))$ aggregated as $F = \bigoplus_{d \in \mathcal{D}} F_d$; (ii) *Deformable convs* (Dai et al., 2017) with offsets/masks $(\Delta p, m) = \phi(F)$, producing $\boldsymbol{Y}' = \mathcal{C}_{\mathrm{def}}(F; \Delta p, m)$ to adapt to articulation; (iii) *Angle-sensitive pooling* $P = \bigoplus_{\theta \in \Theta} \mathcal{P}(\mathcal{R}_\theta(\boldsymbol{Y}'))$ to encode orientation cues. A linear head maps $P$ to joints, $\hat{\boldsymbol{a}} = \mathrm{MLP}(P)$.

**Objective and gains.** We minimize a weighted smooth-$\ell_1$ objective with per-joint weights reflecting range heterogeneity:

$$\mathcal{L}(\delta) = \mathbb{E}_{(\boldsymbol{x}, \boldsymbol{a}) \sim \mathcal{D}_{\mathrm{agnostic}}} \, d\big(\hat{\boldsymbol{a}}(\boldsymbol{x}; \delta), \, \boldsymbol{a}\big).$$

Empirically, arm decoupling improves action prediction by $\sim 20\%$ over a monolithic baseline, and DAD adds a further $\sim 20\%$, meeting the 0.06 precision required for video-driven manipulation replay.

### 3.3.2 COUPLING WITH TASK SEMANTICS

For accomplishing manipulation tasks, a straightforward approach is to build a model pipeline with a video generation model $\mathcal{M}_x : \mathcal{L} \times \mathcal{X} \to \mathcal{X}^N$ and an inverse dynamics model $\mathcal{M}_a : \mathcal{X} \to \mathcal{A}$. Here AnyPos ($\mathcal{F}_\delta$) serves as the IDM ($\mathcal{M}_a$), mapping given observations into actions. At inference, the visual generation model $\mathcal{M}_x(\boldsymbol{x}_T, \ell)$ generates task-aligned futures $\boldsymbol{x}_{T+1:T+k}$ from the current observation $\boldsymbol{x}_T$ and instruction $\ell$. The IDM $\mathcal{M}_a(\boldsymbol{x}_T)$ then maps each predicted frame to an action, giving a sequence of actions $\boldsymbol{a}_{T+1:T+k}$. This modular design keeps data efficiency, enables zero-shot or few-shot transfer by updating only $\mathcal{M}_x$, and cleanly separates image-space planning from low-level feasibility via $\mathcal{F}_\delta$.

## 4 EXPERIMENTS

To evaluate whether AnyPos has learned a good feasible action and embodiment modeling prior from the task-agnostic dataset $\mathcal{D}_{\mathrm{agnostic}}$, and how it enhances task-specific models, we conduct three progressively rigorous tests: (a) Action Prediction Accuracy: We compare the performance of AnyPos against standard baselines (ResNet, which is used in (Du et al., 2023; Yang et al., 2024; Zhou et al., 2024; Black et al., 2023)), and task-specific datasets) on a unified test benchmark to assess its high-precision action prediction capability. (b) Real-World Replay: We test the robustness of AnyPos on common and unseen long-horizon tasks by executing its predictions through ground-truth videos, comparing success rates with baselines. (c) Real-World Model-Pipeline Deployment: Coupling with other models (e.g., video generation models), AnyPos consistently completes diverse tasks using generated (non-real) video inputs.

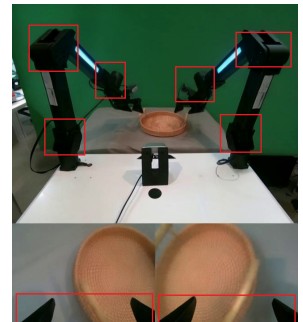

Figure 3: The schematic of the dual-arm setup. The red box is added manually, not model input. The bottom-left/right subfigures display left/right grippers. The top subfigure depicts the 2 lightweight 6-DOF robotic arms, each comprising 2 base joints, 1 elbow joint, and 3 high-precision wrist joints.

### 4.1 EXPERIMENTAL SETUP

**Real Robot:** Mobile ALOHA (Fu et al., 2024) is a commonly used mobile dual-arm robot for manipulation tasks. Each 6-DoF arm has a gripper, creating a 14-dimensional action space for various tasks. We modify it with three RGB cameras: two wrist-mounted and one rear-mounted elevated camera to observe the workspace. This setup provides complete visual data for IDMs' qpos predictions. The model uses this input to predict all 14 joint positions for robot position control. The red box in Fig. 3 (added manually, not part of model input) emphasizes the wrist joint details, which are crucial for high-precision tasks.

**Training Dataset:** We collect 610k task-agnostic image-action pairs, along with human-teleoperation training data for comparison. AnyPos's task-agnostic action coverage across all action dimensions in the test dataset, demonstrating the comprehensiveness of our data-collection methods. (see Appendix A.4).

**Evaluation Method:** We evaluate prediction accuracy (Sec. 4.2) using 13 teleoperated manipulation tasks (2.5k image-action pairs) with unseen skills/objects. For real-world tasks, we assess AnyPos's

success rate with ground-truth videos (Sec. 4.4) and demonstrate 14 tasks with AI-generated videos (Sec. 4.5).

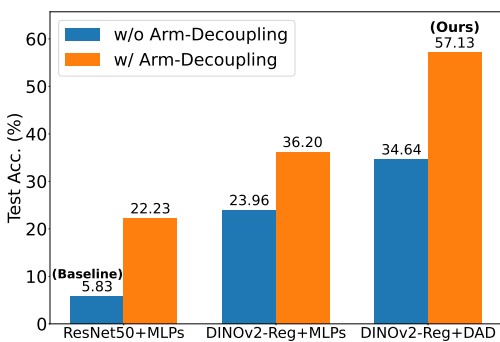

(a) Accuracy on Manipulation Test Dataset.

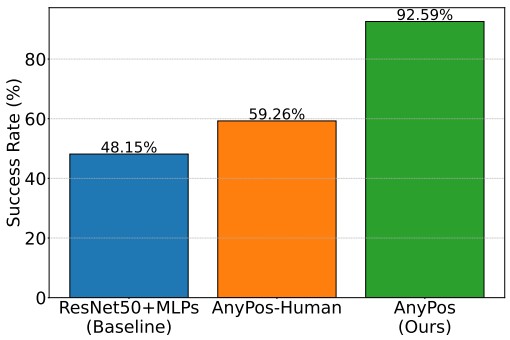

(b) The Success Rates Benchmark of Video Replay.

Figure 4: (a) The Accuracy Benchmark on Manipulation Test Dataset. All the models are trained on the 610k task-agnostic AnyPos dataset. We only report the test accuracy as the predictions of the models are deterministic. (b) The Success Rates Benchmark of Video Replay. Refer to Appendix A.7 for specific task demonstrations and statistical information. AnyPos-Human is trained on data collected from humans, whereas other models are trained on task-agnostic AnyPos data.

## 4.2 FULL EVALUATION OF TASK-AGNOSTIC DATA

Table 1: The Comparison of Human Data (human-collected manipulation data) and AnyPos (Task-Agnostic Actions) method. SR denotes the success rate.

|  | Test Acc. | Replay SR | Collection Time | Dataset Size | Manpower? |
|---|---|---|---|---|---|
| Human Data | **57.78%** | 59.26% | ∼ 2 days (16h) | 33k | Yes |
| AnyPos | **57.13%** | **92.59%** | **∼ 10h** | **610k** | **Automatic** |

To fully assess AnyPos's data collection framework's potential, we evaluate it across three critical dimensions: data quality, collection efficiency, and labor requirements.

For comparison, we collect a human-teleoperated training dataset with 33k image-action pairs of manipulation tasks. This data collection process is labor-intensive and time-consuming, taking 2 days to complete. In comparison, it only took 10 hours for AnyPos to collect 610k task-agnostic image-action pairs without human labor, speeding up data collection by $30\times$.

We evaluate AnyPos trained on the task-agnostic dataset and that trained on the human-collected dataset on two experimental tasks: namely, action prediction accuracy experiment and real-world replay experiment. Detailed descriptions of action prediction accuracy experiment and real-world replay experiment can be found in Sec. 4.3 and Sec. 4.4, respectively.

As shown in Tab. 1, AnyPos trained on the 610k AnyPos dataset matches the test accuracy of the human-collected test dataset. In comparison, Fig. 4b, AnyPos trained with AnyPos dataset outperform that trained on human-collected dataset in real-world replay tasks. The demonstrated high data quality of the AnyPos dataset is primarily due to the uniform spatial distribution of robot positions in the workspace.

## 4.3 EVALUATION OF THE DESIGN OF ANYPOS MODELING

We conduct an action prediction accuracy experiment to test the importance of individual modules and evaluate AnyPos's action prediction accuracy under real-world manipulation task distributions.

For this experiment, we collect human demonstrations of image-action pairs and build a test benchmark with 2.5k samples. Performance is measured as the success rate of predictions where the error

falls below a threshold of 0.06 (except for the gripper, which allows 0.5). This threshold of joint position prediction accuracy was selected through empirical error analysis.

Specifically, AnyPos is compared against two baselines: a widely used ResNet (He et al., 2016)+MLP for embodiment modeling (e.g., IDMs for (Du et al., 2023; Yang et al., 2024; Zhou et al., 2024; Black et al., 2023)), and a DINOv2-Reg (Oquab et al., 2024; Darcet et al., 2024)+MLP model, respectively. We also compare their performance with and without Arm-Decoupled Estimation to assess the decoupling design. Details of model configuration can be found in Appendix B.3.

As shown in Figure. 4a, our AnyPos (i.e., DINOv2-Reg + DAD, enhanced by Arm-Decoupled Estimation), trained on task-agnostic AnyPos data, significantly outperforms other approaches. The Arm-Decoupled Estimation alone improves accuracy by about 20%, while DAD further boosts it by about 21%. Compared to the simple ResNet + MLP used in (Du et al., 2023; Yang et al., 2024; Zhou et al., 2024; Black et al., 2023), our method achieves a 56% higher accuracy.

Table 2: The Comparison of GPTDecoder, DiffusionDecoder and Direction Aware Decoder (AnyPos) as action decoder, with DINO-Reg as vision encoder. SR denotes the success rate.

|  | Parameters | RoboTwin SR | Test Acc. |
| --- | --- | --- | --- |
| GPTDecoder | 118.9M | 48.67% | 19.43% |
| DiffusionDecoder | 90.3M | 58.78% | 35.25% |
| Direction Aware Decoder (AnyPos) | 89.5M | **70.72%** | **57.13%** |

To further evaluate the effectiveness of our Direction Aware Decoder, we conduct an ablation study comparing it with two other decoders: the GPTDecoder and the DiffusionDecoder. Both are policy heads adopted from RoboFlamingo (Li et al., 2024b), a prominent VLA model that combines a visual language model with interchangeable action decoders. Specifically, we adopt DINOv2 with register as the vision encoder, as it proved to be the most effective in our earlier evaluation. All models are trained and tested on our human demonstration dataset. In addition, we introduce a new training set from the RoboTwin 2.0 clean environment (50 tasks, 20 trajectories per task) and a test set from the randomized environment. The training configuration for the decoder remains the same, and the testing setup is consistent with Appendix A.1. As shown in Tab. 2, our model outperforms the other two decoders, reflecting the high quality of embodiment modeling in AnyPos.

The results highlight that AnyPos achieves a significantly higher accuracy in high-precision action prediction compared to other embodiment modeling methods.

## 4.4 EVALUATION OF REAL-WORLD REPLAY

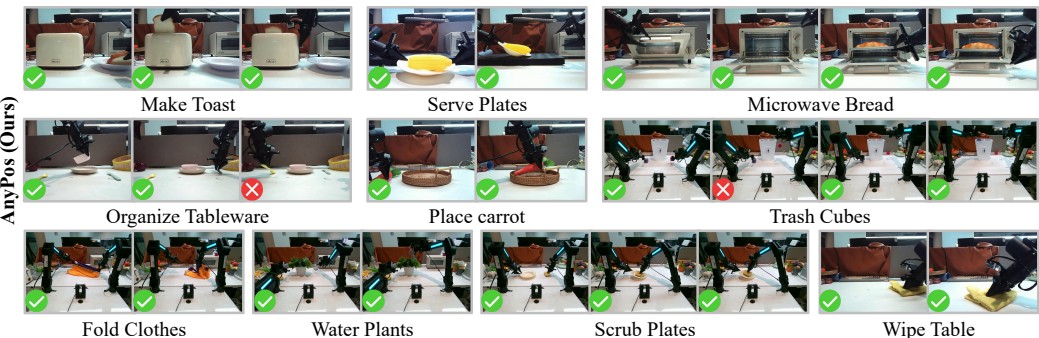

Figure 5: The results of AnyPos with video replay to accomplish various manipulation tasks.

To further test the embodiment modeling ability of AnyPos, we conducted a series of long-horizon, high-precision replay experiments in real-world setting. First, human operators record robot-view videos of teleoperated task executions. The environment is then reset to the initial state shown in the video. Next, we feed each frame of these ground truth videos to the IDMs, execute the generated actions, and observe whether the robot completes the tasks successfully under the same initial conditions.

Our real-robot replay tasks consist of 10 bimanual tasks across 18 objects. Each manipulation task consists of multiple finer sub-steps to evaluate the stability of AnyPos in long-horizon execution.

Fig. 5 and Fig. 4b show AnyPos significantly outperforming both the ResNet50 baseline (+44.4%) and AnyPos-Human (trained on human data) (+33.3%) in replay tests, completing nearly 100% of task steps. Failures primarily occur in highly specific corner cases, falling into two distinct categories. One category involves reset errors. For example, in the Organize Tableware task, a minor fork misalignment during environment reset can cause the gripper to miss the fork during execution and thus result in failure. The other category involves limited error tolerance in teleoperation data. For example, in the Trash Cubes task, human operators sometimes placed cubes too close to the trash bin's rim while attempting to trash it, causing unexpected dislodgement during robotic replay when the cube tripped over the rim in the trash attempt. Despite only 57% action prediction accuracy, AnyPos achieves high real-world success because few critical actions need high precision, while others are more forgiving. Experiments demonstrate that AnyPos reliably reproduces human behaviors from the replay video.

These results show that even 610k steps of automated random action collection (collected in 10 hours) can effectively enable AnyPos to generalize across diverse and long-horizon manipulation tasks.

## 4.5    MODEL-PIPELINE DEPLOYMENT

**Real-World Deployment.**    To evaluate the potential of AnyPos for action prediction and the ability of AnyPos combined with task-specific policies (e.g., video generation models, VLAs, world models) in real-world manipulation tasks, we finetune video generation models (e.g., Vidu (Bao et al., 2024), Wan2.2 (Wan et al., 2025)), following Vidar (Feng et al., 2025) (see Appendix B.5), and combine its outputs with IDM predictions. The video model takes the current RGB observation and generates predicted future observations. AnyPos then processes each video frame to infer actions, which the robot executes. We implement VPP (Hu et al., 2024) as our baseline, following their approach of coupling a video generation model with an action diffusion model. For fair comparison, we use the same fine-tuned video generation model as in our main pipeline (VGM+IDM).

As shown in Fig. 13, our AnyPos, when combined with video generation models, can successfully complete real-world tasks, such as lifting the basket, clicking, and picking up and placing various objects, even when the generated videos are non-real and slightly blurred (Appendix A.8). This demonstrates the potential of integrating AnyPos with generated videos for real-world manipulation tasks.

Table 3: The Success Rates Benchmark of Real-World Experiments.

| Tasks | VGM+AnyPos (Ours) | VPP (Hu et al., 2024) |
|---|---|---|
| Placing bread into steam baskets | **100%** | 0% |
| Transferring apples to fruit baskets | **60%** | 0% |
| Wiping tables with rags | **60%** | 40% |

To further test the background generalization of AnyPos in real-world environment, we conduct extended experiments (placing bread into steam baskets, transferring apples to fruit baskets, and wiping tables with rags), all performed against complex, unseen physical backdrops. Our VGM+AnyPos framework achieved success rates of 100%, 60%, and 60% in the three experiments respectively. Primary failures stemmed from inherent limitations in video generation precision.

**Simulation Benchmarking.**    Additionally, Tab. 4 provides a comprehensive comparison with leading baseline models on the robotwin benchmark. We trained a single, masked (Feng et al., 2025) AnyPos model across all tasks, using 20 clean-environment demonstrations per task. The baselines were obtained from the official RoboTwin 2.0 Chen et al. (2025) leaderboard. They follow a per-task training scheme, with a separate model trained for each task, each utilizing 50 trajectories on clean environment. All the models are evaluated in the same clean environment. As shown in the 17 manipulation tasks, our method (AnyPos), when combined with high-level policies like video generation models, achieves strong performance. It surpasses the previous state-of-the-art methods, RDT and Pi0, by **34%** and **23%** in average success rate, respectively. Notably, our model is trained across multiple tasks within a single model, whereas the baseline models are trained individually for each task, which further highlights our model's stable performance across tasks.

Table 4: Success Rates of 17 Tasks in RoboTwin 2.0.

| Task / Success Rate (%) | AnyPos(Ours) | RDT | Pi0 | ACT | DP | DP3 |
|---|---|---|---|---|---|---|
| Adjust Bottle | 95 | 81 | 90 | 97 | 97 | **99** |
| Click Alarmclock | **100** | 61 | 63 | 32 | 61 | 77 |
| Click Bell | **95** | 80 | 44 | 58 | 54 | 90 |
| Grab Roller | **100** | 74 | 96 | 94 | 98 | 98 |
| Lift Pot | 75 | 72 | 84 | 88 | 39 | **97** |
| Move Can Pot | 50 | 25 | 58 | 22 | 39 | **70** |
| Move Pillbottle Pad | **70** | 8 | 21 | 0 | 1 | 41 |
| Move Playingcard Away | **100** | 43 | 53 | 36 | 47 | 68 |
| Pick Dual Bottles | **75** | 42 | 57 | 31 | 24 | 60 |
| Place Container Plate | **100** | 78 | 88 | 72 | 41 | 86 |
| Place Empty Cup | **100** | 56 | 37 | 61 | 37 | 65 |
| Place Object Stand | **95** | 15 | 36 | 1 | 22 | 60 |
| Press Stapler | **90** | 41 | 62 | 31 | 6 | 69 |
| Shake Bottle | **100** | 74 | 97 | 74 | 65 | 98 |
| Shake Bottle Horizontally | **100** | 84 | 99 | 63 | 59 | **100** |
| Stack Bowls two | 85 | 76 | **91** | 82 | 61 | 83 |
| Turn Switch | **70** | 35 | 27 | 5 | 36 | 46 |
| Average Success Rate | **88.24** | 55.59 | 64.88 | 49.82 | 46.29 | 76.88 |

We provide additional ablation studies on RoboTwin in Appendix A.1, evaluating AnyPos's performance under challenging visual conditions such as partial occlusion of the robotic arm or when it moves out of the camera view. Our experiments demonstrate that the model remains robust even when the arm exits the view, as critical grasping actions are consistently performed within the visible frame. We also compare task success rates using ground truth videos versus videos generated by the VGM pipeline. Results indicate that ground truth video+AnyPos achieves a marginally higher success rate than VGM+AnyPos, suggesting that the actions predicted by the IDM are sufficient for near-perfect execution and that AnyPos's own error is effectively negligible. These findings are presented in Appendix A.1.

## 5 DISCUSSIONS

This work formally introduces task-agnostic actions for embodiment modeling, demonstrating their potential for general-purpose embodied manipulation and their advantages over task-specific actions in terms of efficiency, cost-effectiveness, and performance. Our whole method introduces 2 components: (1) Task-agnostic Data: Efficiently and scalably collecting task-agnostic random actions to mitigate action data scarcity in embodied AI, (2) Model trained with task-agnostic Data: AnyPos with Arm-Decoupled Estimation and Direction-Aware Decoder to effectively and robustly predict high-precision actions. Experiments demonstrate that AnyPos significantly outperforms previous methods in action prediction accuracy (**+51%**) and real-world dual-arm manipulation success rates (+30∼40%). Additionally, we validate the synergistic potential of AnyPos combined with task-specific policies (e.g., video generation models) in both simulation and real-world manipulation tasks.

**Limitation and Discussion** Replay tasks requiring fine manipulation (e.g., tying knots, laptop power adapter connection) were excluded because human operators could not collect reliable tele-operation data, and real-world model-pipeline deployment is still limited by the capabilities of current video generation models. Furthermore, for each embodiment or altered camera viewpoint, AnyPos must first collect task-agnostic action data for embodiment modeling and establishing a prior for feasible actions specific to that embodiment. These factors prevent us from fully testing and leveraging AnyPos's potential. In addition, we will improve background generalization, enhance the task-agnostic dataset, and expand the action space to support multiple robotic platforms and dynamic manipulation. This will enable AnyPos to serve as an adapter between general embodied models and robot-specific actions.

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

# A MORE RESULTS

## A.1 FURTHER STUDY ON ROBOTWIN

In the RoboTwin environment, we evaluate how partial occlusion of the robot arm and video prediction error affect the performance of the video+AnyPos pipeline, respectively. Both models are trained across multiple tasks.

**Partial occlusion scenarios.** We follow the leaderboard of RoboTwin to collect data for fine-tuning the video generation model and training AnyPos. To be more specific, we collect 50 episodes per task under the clean scenario on RoboTwin using the original camera viewpoints, where partial arm occlusion frequently occurs. We finetune Wan2.2 (Wan et al., 2025) following Vidar(Feng et al., 2025) as our video generation model.

**Error propagation analyses.** We directly use the collected ground truth task completion videos as video input for AnyPos.

We select 17 tasks and conduct 20 trials for each task. The results are shown in Tab. 5. AnyPos-occ denotes the occluded case. AnyPos-gt denotes the error propagation analyses case, where we use the ground truth video instead of generated videos as the input.

Our experiments show that AnyPos is robust to the arm exiting the view, as the critical grasping actions are consistently performed within the visible frame. Moreover, AnyPos achieves a comparable success rate distribution with real videos, suggesting that the error attributable to the IDM is negligible.

Table 5: Success Rates of 17 Tasks in RoboTwin

| Task / Success Rate (%) | AnyPos | AnyPos-occ | AnyPos-gt |
|---|---|---|---|
| Adjust Bottle | 95 | 70 | 100 |
| Click Alarmclock | 100 | 100 | 100 |
| Click Bell | 95 | 100 | 100 |
| Grab Roller | 100 | 95 | 100 |
| Lift Pot | 75 | 100 | 100 |
| Move Can Pot | 50 | 75 | 90 |
| Move Pillbottle Pad | 70 | 80 | 100 |
| Move Playingcard Away | 100 | 95 | 100 |
| Pick Dual Bottles | 75 | 80 | 100 |
| Place Container Plate | 100 | 95 | 95 |
| Place Empty Cup | 100 | 85 | 100 |
| Place Object Stand | 95 | 80 | 100 |
| Press Stapler | 90 | 100 | 100 |
| Shake Bottle | 100 | 100 | 100 |
| Shake Bottle Horizontally | 100 | 100 | 95 |
| Stack Bowls two | 85 | 100 | 90 |
| Turn Switch | 70 | 50 | 80 |
| Average Success Rate | 88.24 | 88.53 | 97.06 |

## A.2 DEMONSTRATION OF CROSS-ARM INTERFERENCE

To investigate potential interference between the two arms during IDM inference, we visualize the attention maps derived from input image gradients. Our analysis reveals that even when estimating the qpos of a single arm, the other arm still receives significant attention, demonstrating the presence of cross-arm interference in the model's processing.

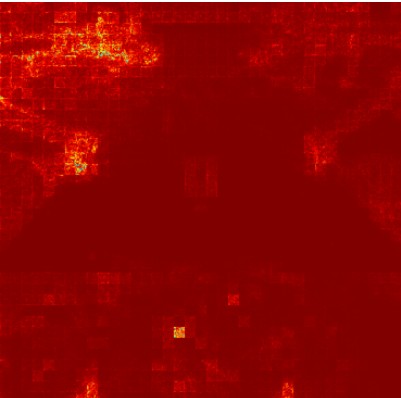

Figure 6: **Attention heatmap of the input image.** Here we only estimate the qpos of the left arm, but there is a clear attention focus on the right arm. demonstrating that the model can not fully distangle the two arm during inference.

### A.3  ANALYSIS OF EXPLORATION EFFICIENCY AND SAFETY

This section provides a qualitative analysis comparing our AnyPos data collection framework against a naive random action collection baseline. Fig. 7 reveals three fundamental limitations in naïve task-agnostic data collection, namely inefficient coverage of reachable states, redundant or degenerate motions (e.g., arms exiting the field of view), and frequent self-collisions. Our AnyPos data collection framework systematically addresses each limitation through its automated, task-agnostic design, enabling dense coverage, diverse behavior generation, and built-in safety mechanisms.

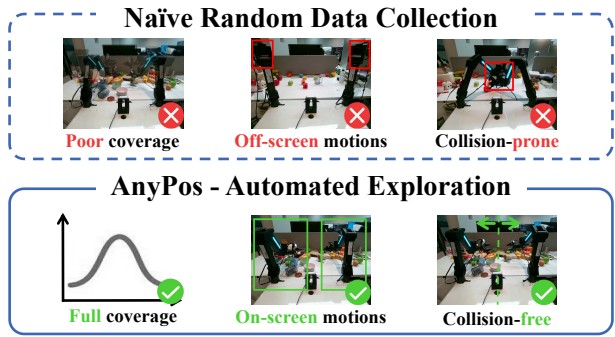

Figure 7: **Visual comparison between naive random action collection (upper) and our proposed AnyPos framework (lower).** Here we highlight three key limitations in the baseline approach: (a) inefficient coverage, (b) redundant motions, and (c) self-collisions. Our method demonstrates superior coverage density, in-frame behavior generation, and inherent safety constraints.

### A.4  DISTRIBUTION OF TASK-AGNOSTIC ANYPOS DATASET AND TEST DATASET

To measure the coverage of the action space of our random actions, we evaluate the distribution of qpos on each dimension, shown in Figure 8.

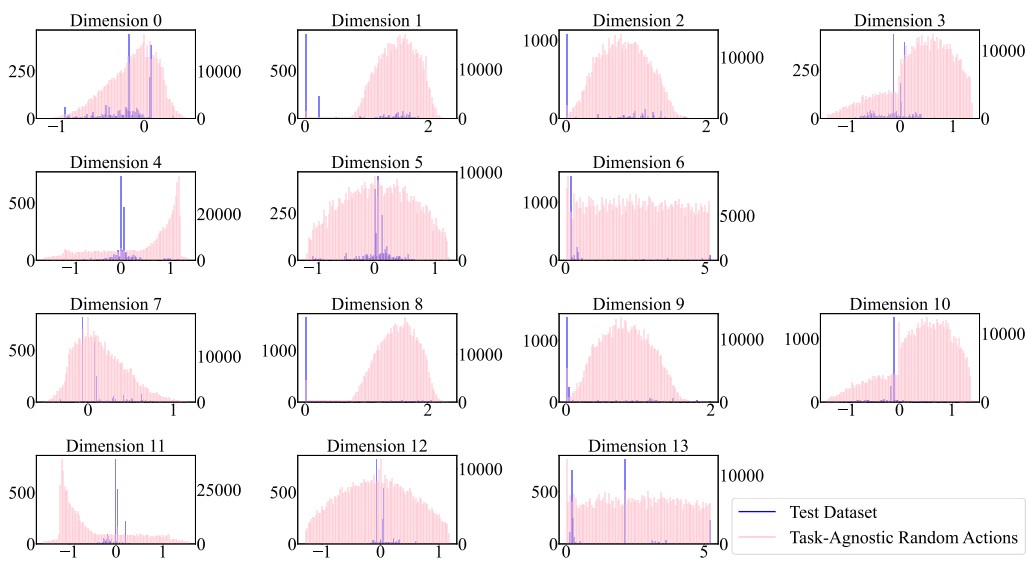

Figure 8: **Qpos distribution of task-agnostic random actions and test dataset**. The figure calculates the frequency distribution of qpos in 14 dimensions. We show that random action can cover all the possible qpos in each dimension. Note that the volume of task-agnostic data significantly exceeds that of the test dataset.

## A.5   DATA-SCALING ANALYSIS

We studied the scaling laws governing our method, quantifying its performance improvement with increasing volumes of training data.

In practice, we trained the model on subsets of the full dataset, ranging from 50K to 610K image-action pairs. We keep the training steps proportional to the size of the dataset.

The results, visualized in Figure 9, reveal a logarithmic growth trend in accuracy as the dataset scales up. This scaling behavior indicates that our method consistently benefits from additional training data, providing valuable guidance for practical applications where data collection costs must be balanced against performance requirements.

Additionally, real-world robot accuracy reached **92.59%** when test set accuracy is only **57.13%**, underscoring the practical scalability of our model.

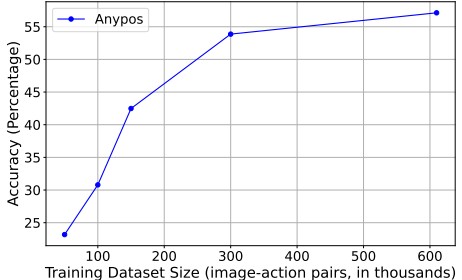

Figure 9: The accuracy of AnyPos training on dataset with different size.

## A.6 EVALUATION OF ACTION PREDICTION

The results presented in Table 6 demonstrate the performance of various methods on the Manipulation Test Dataset. We compare the performance of DINOv2 against ResNet50, MLPs with DAD, with and without Arm-Decoupling, and task-agnostic data versus human data.

Table 6: The Test Accuracy and Error Benchmark on Manipulation Test Dataset. Due to the gripper's higher tolerance for errors, the gripper's error significantly impacts the overall error. Therefore, the Test L1 Error in the table is calculated after excluding the gripper.

| Methods | Arm-Decoupling? | Data | Test Acc. | Test L1 Error |
|---|---|---|---|---|
| ResNet50 + MLPs | No | Task-agnostic Data | 5.83% | 0.1022 |
| DINOv2-Reg + MLPs | No | Task-agnostic Data | 23.96% | 0.0440 |
| DINOv2-Reg + DAD | No | Task-agnostic Data | 34.64% | 0.0491 |
| ResNet50 + MLPs | Yes | Task-agnostic Data | 22.23% | 0.0444 |
| DINOv2-Reg + MLPs | Yes | Task-agnostic Data | 36.20% | 0.0352 |
| **DINOv2-Reg + DAD** | **Yes** | **Task-agnostic Data** | **57.13%** | **0.0282** |
| DINOv2-Reg + DAD | Yes | Human Data | **57.78%** | **0.0203** |

## A.7 EVALUATION OF REAL-WORLD VIDEO REPLAY

Fig. 10, Fig. 11, and Fig. 12 show the detailed replay performance of AnyPos, baseline (ResNet+MLP), and AnyPos-Human (trained with human-collected data) on the manually collected real-world video replay dataset, respectively.

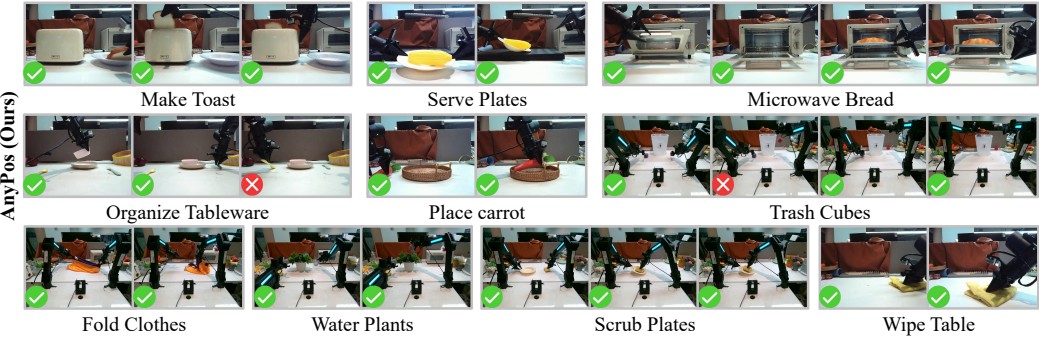

Figure 10: The results of AnyPos collaborating with video replay to accomplish various manipulation tasks.

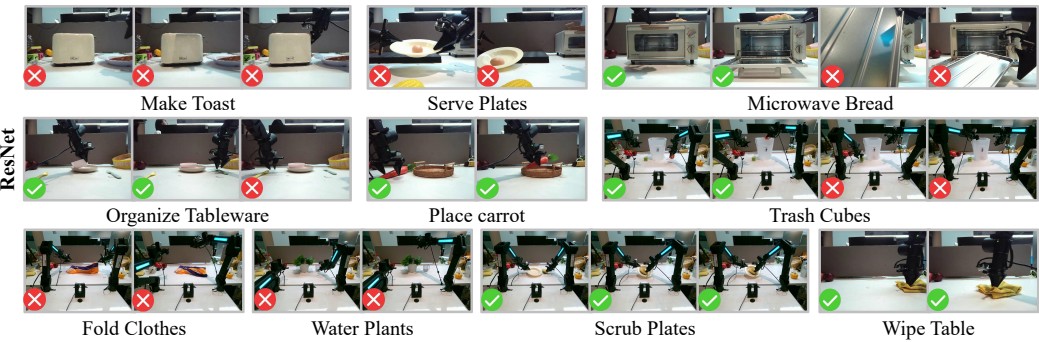

Figure 11: The results of baseline (ResNet+MLP) collaborating with video replay to accomplish various manipulation tasks.

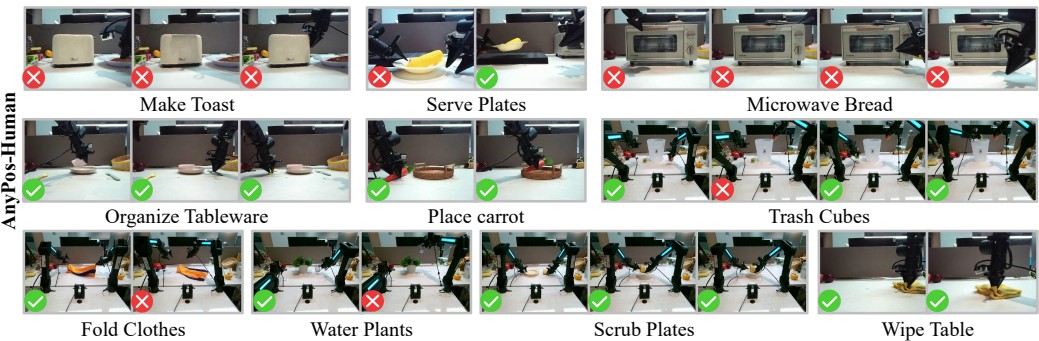

Figure 12: The results of AnyPos-Human (trained with human-collected data) collaborating with video replay to accomplish various manipulation tasks.

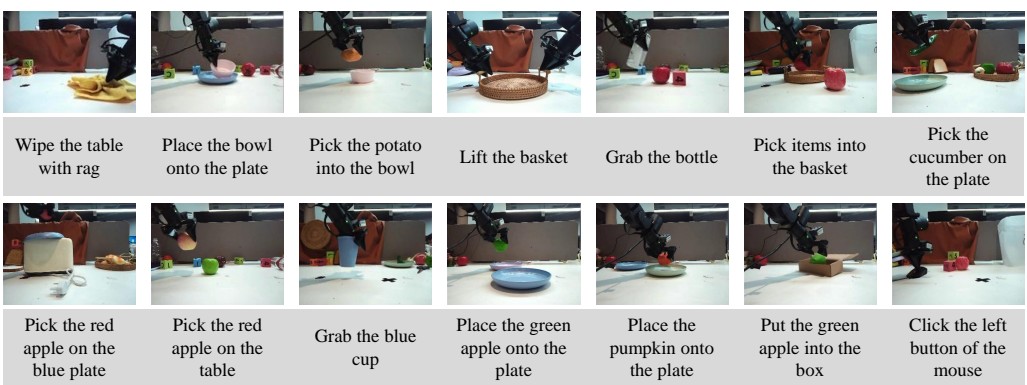

Figure 13: The results of AnyPos collaborating with video generation models to accomplish various manipulation tasks.

### A.8   REAL-WORLD DEPLOYMENT WITH VIDEO GENERATION MODEL

Fig. 14 demonstrates how AnyPos collaborates with a video generation model in real-world deployment. Especially when the robotic arm is a bit blurry in the generated video, AnyPos can still complete the manipulation task. More detailed execution videos can be found in the supplementary materials.

## B   IMPLEMENTATION DETAILS

### B.1   ANYPOS DATASET AND PPO IMPLEMENTATION

Our PPO implementation is built on rsl_rl. Key settings of PPO and AnyPos Dataset are summarized in Table 7.

### B.2   REWARD FUNCTION

To ensure the policy in the AnyPos dataset collection achieve the desired behavior on our robot, we design a reward function that reflects the task's objectives. We design a multi-stage reward function focusing on EEF goal distances, action rate and joint velocity, in order to yield higher-quality data collection.

Definitions of each part of our reward functions are listed as follows:

1. **EEF Goal Distance**

$$R_{\text{reaching\_obj}} = \left(1 - \tanh\left(\frac{\|\mathbf{p}_{\text{object}} - \mathbf{p}_{\text{ee}}\|_2}{\sigma}\right)\right)$$

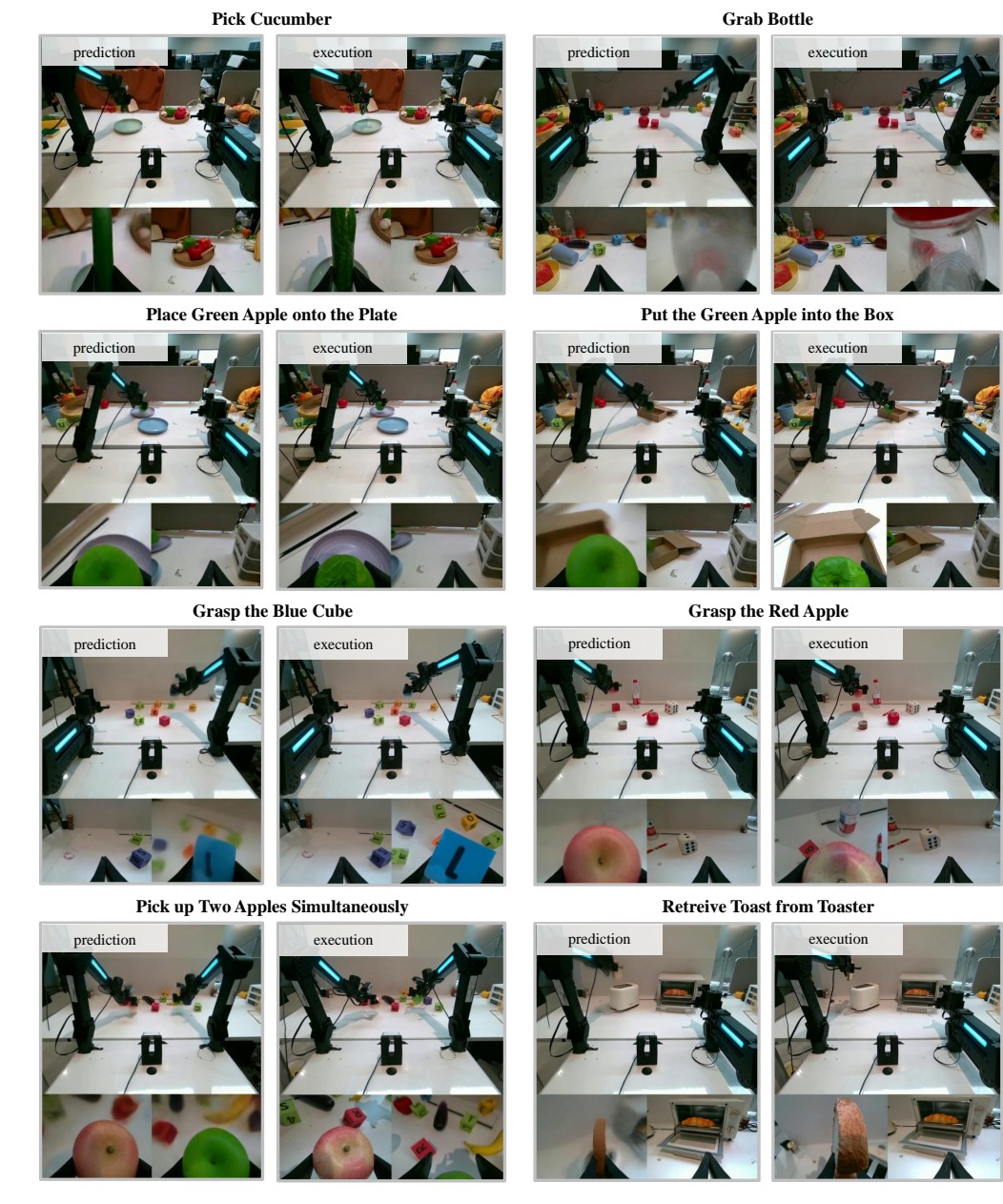

Figure 14: Sampled results of AnyPos collaborating with generated video to accomplish various manipulation tasks. In tasks such as "Grasp the Blue Cube" and "Grab Bottle", the generated video frames on the left exhibit blurred wrist joint details of the robotic arm. Nevertheless, AnyPos successfully accomplishes the manipulation task under these conditions.

where $\mathbf{p}_{\text{object}}$ denotes the target position in world coordinates. $\mathbf{p}_{\text{ee}}$ denotes the position of the end-effector in world coordinates. $\sigma$ is a scaling factor for distance normalization. In this term, $\sigma = 0.08$.

2. **EEF Goal Distance (Fine-Grained)**

$$R_{\text{reaching\_obj\_fine}} = \left(1 - \tanh\left(\frac{\|\mathbf{p}_{\text{object}} - \mathbf{p}_{\text{ee}}\|_2}{\sigma}\right)\right)$$

The formulation is identical to the preceding term, but $\sigma$ is smaller foir finer control. In the term, we let $\sigma = 0.01$.

Table 7: Parameters of PPO and AnyPos Dataset.

| Parameters of PPO | Value |
|---|---|
| Clip Param. of PPO | 0.2 |
| Value Function Clipping | True |
| Value Loss Coeff. | 1.0 |
| Desired KL Divergence | 0.01 |
| Entropy Coef. | 0.01 |
| gamma | 0.98 |
| GAE (lambda) | 0.95 |
| Gradient Clipping | 1.0 |
| Learning Rate | 0.001 |
| Mini-Batch | 4 |
| The Number of Steps per Env per Update | 24 |
| Learning Epochs | 5 |
| Schedule | adaptive |
| Empirical Normalization | True |
| Target EEF position Range of Left Arm | $x \in (0.36, 0.7), y \in (-0.08, 0.41), z \in (0.6, 1.0)$ |
| Hidden Dim. of Actor | [512, 256, 128] |
| Hidden Dim. of Critic | [512, 256, 128] |
| Activation | Elu |

| Parameters of AnyPos Dataset | Value |
|---|---|
| Dataset Size (steps) | 610k image-action pairs |
| Dataset Size (trajectories) | 638 |
| Input | Concatenated image of high, left-wrist, and right-wrist views |
| Image Resolution | 640*720 |
| Output | 14-dim joint position |
| Content | Task-agnostic random dual-arm trajectories collected by AnyPos |
| Virtual Random Boundary Plane $\mathcal{B}$ | $y \in (-0.15, 0.15)$ |
| Target EEF position Range of Left Arm | $x \in (0.36, 0.7), y \in (-0.08, 0.41), z \in (0.6, 1.0)$ |
| Target EEF position Range of Right Arm | $x \in (0.36, 0.7), y \in (-0.41, 0.08), z \in (0.6, 1.0)$ |
| Interval Threshold between Arms | 0.15 |

3. **Action Rate Penalty**

$$R_{\text{act\_rate}} = -\|\mathbf{a}_t - \mathbf{a}_{t-1}\|_2^2$$

where $\mathbf{a}_t$ denotes the action at current time step $t$, while $\mathbf{a}_{t-1}$ denotes the action at the previous time step $t-1$.

4. **Joint Velocity Penalty**

$$R_{\text{joint\_vel}} = -\sum_{i \in \text{joint\_ids}} \dot{q}_i^2$$

where joint_ids denotes the set of joint indices whose velocities are to be penalized, and $\dot{q}_i^2$ is the velocity of the $i$-th joint in the set.

The total reward is the weighted sum of each reward function:

$$\phi_{\text{coll}} = w_{\text{reaching\_obj}} \times R_{\text{reaching\_obj}} + w_{\text{reaching\_obj\_fine}} \times R_{\text{reaching\_obj\_fine}}$$

$$\phi_{\text{limit}} = w_{\text{act\_rate}} \times R_{\text{act\_rate}} + w_{\text{joint\_vel}} \times R_{\text{joint\_vel}}$$

where the weight design for the reward function is: $w_{\text{reaching\_obj}} = 200$, $w_{\text{reaching\_obj\_fine}} = 100$, $w_{\text{act\_rate}} = -1 \times 10^{-4}$, and $w_{\text{joint\_vel}} = -1 \times 10^{-4}$.

B.3    MODEL CONFIGURATION

The model configuration of Anypos and other models trained on task-agnostic action dataset is listed in Table 8. The model accepts 4 images as input, two from the wrist cameras, and two from the front camera divided by the split-line algorithm. The four images are resized to the same size of $518 \times 518$ and normalized.

For training on human-collected data, only replace the iteration to 48000, because human-collected data is smaller, thus the epoch will be larger. The model converges after 48000 iterations on human-collected data (validation accuracy: 97.8%).

Table 8: **Configuration of Different Models Trained on Task-Agnostic Action Dataset**

| | Models | Value |
|---|---|---|
| DINO-Reg | Hidden Size | 768 |
| | Hidden Layers | 12 |
| | Model Size | 86.6M params |
| | Pretrained | Yes |
| MLP-regressor | Convolution | $1 \times 1, (768, 2)$ |
| | MLP | $(2738, 256), (256, 14/6/1)$ |
| | Activation Function | GELU |
| | Model Size | 0.71M params |
| DAD | $\mathcal{D}$ | $\{1, 2, 3, 6\}$ |
| | $\Theta$ | $\{0°, 45°, 90°, 135°\}$ |
| | MLP | $(256, 512), (512, 14/6/1)$ |
| | Activation Function | GELU |
| | Model Size | 2.96M params |
| ResNet50 | Input | $224 \times 224$ |
| | MLP | $(2048, 14/6/1)$ |
| | Model Size | 23.6M params |
| | Pretrained | Yes |
| Training | Batchsize | 8 |
| | Iteration | 96000 |
| | Optimizer | AdamW, $\beta = (0.9, 0.999), \epsilon = 0.01$ |
| | Learning Rate | $5 \times 10^{-5}$ for DINO-Reg, $5 \times 10^{-4}$ for the rest |
| | Weight Decay | 0.01 |
| | LR Scheduler | Cosine Scheduler |
| | Warmup Steps | 9600 |
| | Weighted Smooth L1 Loss | $: d(x, \hat{x}) = \begin{cases} 0.5\mathbf{w} \cdot \frac{(x-\hat{x})^2}{\beta} & \text{if } \|x - \hat{x}\| < \beta \\ \mathbf{w} \cdot (\|x - \hat{x}\| - 0.5\beta) & \text{otherwise} \end{cases}$ |
| | $\beta$ | 0.1 |
| | $\mathbf{w}$ | $w_{4,11} = 2, w_{\{0,1,\dots,13\}-\{4,11\}} = 1$ |
| Data Augmentation | ColorJitter | Brightness Range: (0.8, 1.2) |
| | | Contrast Range: (0.7, 1.3) |
| | | Saturation: (0.5, 1.5) |
| | | Hue: 0.05 |
| | Randomize Background | Randomize pixels in non-arm-colored background. |
| | | Random Apply Probability: 0.4 |
| | Random Adjust Sharpness | Sharpness Factor: 1.8 |
| | Sharpness Probability | 0.7 |
| | Resize | (518, 518) |
| | Normalization | $mean = [0.485, 0.456, 0.406]$ |
| | | $std = [0.229, 0.224, 0.225]$ |

### B.3.1 ARM-DECOUPLED ESTIMATION TO REDUCE HYPOTHESIS SPACE

Our approach consists of two stages: (1) Arm Segmentation: Leveraging the fact that the pedestal joints remain stable and the robotic arms are uniformly black, we use the pedestal joint pixel as a seed point for flood-fill-based arm segmentation to calculate a split line for the image that divides two arms. However, if the two arms overlap or part of the arm goes out of the picture, which causes the flood-fill algorithm to fail, we fall back to a default bounding box strategy, cropping the left or right 3/5 of the image based on arm position prior. (2) Decoupled qpos Estimation: The segmented left and right arm regions are fed into two independent sub-models, each predicting qpos for their respective arm excluding the gripper. Specifically, Gripper states are estimated separately by two additional sub-models that take only the image of the left or right wrist as input. Therefore, by combining split lines with four specialized sub-models, our method achieves arm-decoupled estimation, significantly improving qpos prediction accuracy compared to entangled bimanual approaches.

Table 9: **Composition of Different Models.**

| Model | Arm-Decoupling | Composition |
|---|---|---|
| DINO + DAD (Anypos) | Yes | ($\times$2 Arms) DINO-Reg + DAD |
| | | ($\times$2 Wrists) DINO-Reg + MLP-regressor |
| | No | DINO-Reg + DAD |
| DINO + MLP | Yes | ($\times$4 Arms & Wrists) DINO-Reg + MLP-regressor |
| | No | DINO-Reg + MLP-regressor |
| ResNet50 + MLP | Yes | ($\times$4 Arms & Wrists) ResNet50 |
| | No | ResNet50 |

### B.4 COMPUTATION RESOURCES

We conduct the training on a machine equipped with 8 * 80GB NVIDIA Hopper series GPUs, utilizing Accelerate (Gugger et al., 2022) and Pytorch (Paszke et al., 2019) for multi-GPU parallelism. AnyPos required 25 hours to train on 610k pairs of data for 96,000 iterations * 8 batch size * 8 GPUs.

### B.5 VIDEO GENERATION MODEL

In practical implementation, we finetune Vidu 2.0(Bao et al., 2024) and Wan2.2 (Wan et al., 2025) following Vidar(Feng et al., 2025) as our video generation model. We collected 750,000 multi-view robotic trajectories from open-source datasets (Agibot, RDT, RoboMind) for Stage-1 fine-tuning. Each image provides three distinct perspectives: top-down, left-side, and right-side views. These images do not necessarily align with AnyPos's input requirements. Subsequently, we performed Stage-2 fine-tuning using 230 task-specific trajectories gathered from our specific robotic platform. For the RobotWin benchmark, we collected 50 tasks, each with 20 trajectories, to apply stage-2 fine-tuning to the video generation model.

## C EXPERIMENTAL DETAILS

### C.1 EVALUATION OF ACTION PREDICTION

The parameters of evaluation of action prediction are shown in Table 10.

Table 10: Parameters of evaluation of action prediction.

| Parameter | Value |
|---|---|
| Training Dataset | 610k Task-Agnostic Data or 33k Human-Collected Data |
| Test Dataset | 2.5k Manipulation Dataset |
| Evaluation Threshold on Test Dataset | for $i = 6, 13, d(a_i, \hat{a}_i) < 0.5$ |
| | others: $d(a_i, \hat{a}_i) < 0.06$ |

### C.2 EVALUATION OF REAL-WORLD VIDEO REPLAY

We design our Real-World Video Replay scenario to replicate the daily workspace setting, which includes a typical white laboratory desk, with cluttered objects on the desk, and several computer monitors in the background. We manually collected 10 long-horizon robot manipulation tasks for real-world video replay, which represent ubiquitous daily household chores. Each task exhibits sequential dependency, where successful completion of subsequent stages directly depends on the preceding stage's achievement.

Our 10 tasks include the following tasks and stages:

- Make Toast: (1) Pick toast from plate, (2) Insert toast into toaster slot, (3) Push down the toasting lever.

- Serve Plates: (1) Grip plate with both hands, (2) Position plate forward on the table.
- Microwave Bread: (1) Open microwave door, (2) Retrieve baking tray with bread, (3) Place baking tray inside microwave, (4) Close microwave door.
- Organize Tableware: (1) Position bowl on plate, (2) Place fork on right side of plate, (3) Place spoon on left side of plate.
- Place carrot: (1) Pick up the carrot, (2) Place the carrot in the basket.
- Trash Cubes: (1) Select cube from right side, (2) Dispose cube in trash bin, (3) Select cube from left side, (4) Dispose cube in trash bin.
- Fold Clothes: (1) Fold pants by waistband and hem, (2) Fold pants using waistband grip.
- Water Plants: (1) Hold water-filled cup, (2) Tilt cup to irrigate plant.
- Scrub Plates: (1) Simultaneously grasp sponge and plate, (2) Scrub plate with leftward sponge motion, (3) Scrub plate with rightward sponge motion.
- Wipe Table: (1) Maintain firm rag grip, (2) Wipe table surface with rag.

Due to the deterministic and costly nature of the replaying experiment, real-world implementations of these experiments are typically limited to a single trial.

### C.3 REAL-WORLD DEPLOYMENT WITH VIDEO GENERATION MODEL

The experimental setup of real-world deployment with a video generation model follows that of the real-world video replay experiment, except that the videos used are different. AnyPos processes the generated video frames to infer actions, which are executed by the ALOHA robot. A task is considered successful if the robot accomplishes it as instructed.

## D HARDWARE DETAILS

Tab. 11 and Fig. 15 show the detailed information of our robot.

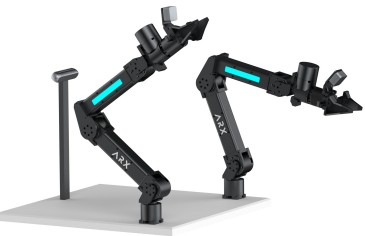

Figure 15: Hardware features.

Table 11: Hardware.

| Parameter | Value |
| --- | --- |
| DoF | $(6 + 1 \text{ (gripper)}) \times 2 = 14$ |
| Size | $770 \times 700 \times 1000$ |
| Arm Weight | 3.9kg |
| Arm Payload | $1500g$ (peak), $1000g$ (valid) |
| Arm Reach | $600mm$ |
| Arm repeatability | $1mm$ |
| Arm working radius | $620mm$ |
| Joint motion range | $J1 : 180° \sim -120°, J2 : 0° \sim 210°$ |
|  | $J3 : -180° \sim 0°, J4 : \pm90°$ |
|  | $J5 : \pm90°, J6 : \pm110°$ |
| Gripper range | $0 \sim 80mm$ |
| Gripper max force | $10N$ |
| Cameras | 3 RGB camears: front$\times$1, wrist$\times$2 |

## E  BROADER IMPACTS

This work advances robotic manipulation by introducing AnyPos, a framework for IDM learning from scalable, task-agnostic action data. The application of this framework in various fields may lead to breakthroughs in automation and intelligent systems, benefiting sectors such as household robotics, healthcare assistance, precision manufacturing, and logistics automation. By reducing reliance on human demonstrations, AnyPos could accelerate the deployment of adaptable robotic solutions in real-world environments.

