# OpenReview forum: "AnyPos: Automated Task-Agnostic Actions for Bimanual Manipulation"
_ICLR.cc/2026/Conference — Submitted to ICLR 2026_

### Official Review · Reviewer_kAye · 2025-10-21

**Soundness:** 3
**Presentation:** 3
**Contribution:** 2
**Rating:** 4
**Confidence:** 5

**Summary:**

The paper proposes AnyPos, a framework for learning robot actions in a task-agnostic way. Its core contribution lies in two connected parts: first, an automated exploration pipeline that uses a reinforcement-trained mapper and safety constraints to sample the robot’s workspace and generate large quantities of collision-free image–action pairs without teleoperation or task labels; second, an inverse-dynamics embodiment model that predicts feasible joint positions from visual inputs using an arm-decoupled design and a direction-aware decoder for sub-degree precision. Together, these components aim to provide a reusable “feasibility prior” that can later be coupled with high-level policy or video-generation models to execute diverse manipulation tasks more efficiently than task-specific datasets.

**Strengths:**

The paper’s strengths lie in its clear problem framing and modular design—it separates physical feasibility learning from task semantics through a well-structured pipeline that’s easy to follow. The authors present a fully automated, high-throughput data collection system that efficiently explores the workspace without teleoperation, coupled with a safety-aware exploration scheme that enforces collision avoidance, joint limits, and inter-arm constraints. The precision-oriented inverse dynamics model introduces practical architectural tweaks, such as arm-decoupled estimation and a direction-aware decoder, which demonstrably improve stability and accuracy. Finally, the real-robot replay results show that the learned embodiment prior can be executed reliably, suggesting the overall framework has strong potential as a practical foundation for scalable, modular robot learning.

**Weaknesses:**

The paper’s main weaknesses lie in its limited empirical validation and unclear justification of core design choices. While the motivation is compelling, the work never actually demonstrates that separating semantics from feasibility improves transfer — all experiments are confined to a single robot, with no cross-task, cross-embodiment, or cross-view evaluations. The decision to use reinforcement learning for deterministic action generation is conceptually weak and lacks comparison against straightforward baselines like inverse kinematics or behavior cloning with physical constraints. The proposed Direction-Aware Decoder adds engineering complexity without clear theoretical grounding or fine-grained ablations to prove necessity. Moreover, the probabilistic decomposition introduced early in the method is never operationalized in training, leaving a gap between the stated formulation and the implemented model. Overall, the paper presents a strong motivation but delivers mostly incremental engineering under a broader conceptual narrative that remains unverified.

**Questions:**

1. Why is reinforcement learning needed for deterministic action mapping, and what concrete advantages does it offer over a straightforward inverse-kinematics or behaviour-cloning approach with safety and collision constraints?
2. Deterministic vs. probabilistic formulation: The paper presents a probabilistic factorization suggesting a distributional treatment of actions, but the actual system predicts a single deterministic joint configuration. How is this formulation relevant if the model never models uncertainty or multiple feasible actions? Since the “world model” is deterministic, how does this help separate “what to do” from “what is physically feasible,” and what additional insight or capability does this probabilistic claim really provide?
3. How does the model handle camera-view and embodiment differences, given that the state representation is trained purely in the 2D image domain—does it rely on multi-view training, 3D understanding, or re-collection for new viewpoints?
4. What evidence supports that separating feasibility from task semantics improves transfer, and can the embodiment model generalize across unseen tasks or robot morphologies without retraining?
5. What is the real contribution of the Direction-Aware Decoder (DAD)? The paper shows marginal experiments but lacks fine-grained ablation; if it is mainly an engineering tweak, why treat it as a core architectural innovation?
6. What concrete metrics and comparisons demonstrate that the RL-generated dataset is superior (in workspace coverage, safety rate, or diversity) to simpler IK-based or rule-based data generation pipelines?

---

> ### Author Response · Authors · 2025-11-23
>
> Thank you for your time and positive feedback! Please find our response below.
>
> ## 1. The Choice of RL over IK/BC
> We clarify that our framework is compatible with both RL and IK for the deterministic mapping. We chose to use RL in our framework because:
> - Our action mapping derives joint positions from only end-effector positions.
> - Behavioral Cloning (BC) is not applicable for this mapping due to the lack of expert demonstration data.
> - Inverse Kinematics (IK) offers two common approaches: some IK control algorithms require both end-effector position and rotation to compute joint positions, while others can determine a feasible set of joint positions using only the end-effector position. However, even with the latter approach, the resulting joint positions, while mathematically valid, may not always be **physically feasible**. This infeasibility can arise from factors such as **workspace limits, joint constraints, self-collision, environmental collision, and kinematic singularities**.
> - In contrast, reinforcement learning (RL) policies directly explore feasible joint positions within the workspace. By incorporating considerations such as termination conditions and reward functions, RL naturally accounts for workspace boundaries, joint limits, collisions, and singularities. As a result, the action mapping obtained through RL is guaranteed to be feasible.
>
> ## 2. Deterministic vs. probabilistic formulation
> We sincerely apologize for the confusion caused by our presentation of the probabilistic formulation. The formulas were intended to formally describe our core motivation: to decompose the robot manipulation problem into a probabilistic "what to do" component (inferred from instructions and observations) and a deterministic "what is physically feasible" component (constrained by the robot's embodiment). We wish to clarify that AnyPos is not a "world model" but an embodiment modeling method responsible for the latter, deterministic part—mapping intents to physically feasible actions. We acknowledge that the current exposition of this concept was unclear and we will undertake a thorough revision of the manuscript to significantly enhance its clarity and readability.
>
> ## 3. Handling Camera-View and Embodiment Differences
> We thank the reviewer for this question. Our approach handles camera-view and embodiment differences by retraining the AnyPos model from scratch for new configurations. This reflects our core design philosophy: we view AnyPos as a lightweight "head" responsible for cross-embodiment transfer. This process, which includes both automated data collection and model training, is highly efficient and can be completed in a few hours (e.g., 5~10 hours + 1 day for our setup). This demonstrates the practicality of our method in adapting to different camera viewpoints and robot embodiments without relying on multi-view training or explicit 3D understanding, as the rapid retraining capability allows for quick deployment in varied settings.
>
> ## 4. Evidence for Separation Benifits and Generalization
> We thank the reviewer for the constructive question.
> 1. **Seperation improves transfer**: To evaluate how this seperating approach improves transfer, we combined our AnyPos with a video generation model and evaluated this pipeline of the RobotWin benchmark (App. A.1, Tab. 3). Quantitive results showed that, when evaluated on 17 tasks in the clean setting, this decoupling approach outperforms SOTA methods like RDT and Pi0 by 32% and 23% respectively.
> 2. **Embodiment model generalization ability**: The embodiment model can generalize across unseen tasks without retraining. However, retraining is required for unseen robot morphologies, as our embodiment model is embodiment-specific. Despite the need for retraining,  it is a quick process. A full retraining cycle from scratch, including automated data collection and model training, takes just 5~10 hours for data collection and approximately a day for model training.
>
> ## 5. Contribution of the Direction-Aware Decoder (DAD)
> We thank the reviewer for bringing up this critical question regarding the DAD's contribution. We acknowledge that ablating DAD against a standard MLP (Fig. 4a) is not enough for demonstrating its contribution to the model and that a more fine-grained ablation would strengthen this claim. We plan to conduct experiments ablating the decoder against other widely used action decoder designs to conclusively demonstrate how this component is fundamental to the model's performance. We will update the results once the experiments are completed.

---

> ### Author Response · Authors · 2025-12-02
> **Question 5: More Comparisons of Action Decoders**
>
> We sincerely thank the reviewers once again for their insightful questions and constructive feedback. We would like to restate the core abstract idea of our paper: AnyPos consists of two stages: **task-agnostic exploration** and **embodiment modeling**. Task-agnostic exploration is designed to discover “what is physically feasible and consistent,” while embodiment modeling learns all feasible actions, thereby decoupling it from the high-level “how to achieve a goal” question. This decoupling allows both high-level goal-conditioned models and AnyPos to be supervised with ample data in isolation. The IDM (visual encoder + action decoder) serves as one way to validate the effectiveness of embodiment modeling. In the future, we expect that task-agnostic embodiment modeling will help improve the generalization capabilities of models such as VLAs and world models.
> Here, we provide further comparison between our proposed Direction Aware Decoder and two types of action decoders used in VLAs.
>
> **More Comparisons of Action Decoders (II. Embodiment Modeling in Fig.1)**: We further conduct an experiment on the action decoder to evaluate the effectiveness of our Direction Aware Decoder—compared to other decoders—in predicting actions based on visual features, which reflects the quality of embodiment modeling in AnyPos. RoboFlamingo [1] is a well-known vision‑language‑action model composed of a VLM combined with different action decoders (policy heads). We select the **GPTDecoder and DiffusionDecoder from RoboFlamingo as our baselines**. Specifically, we adopt the strongest visual encoder identified in our experiments—DINOv2 with register—and compare three action decoders: the Direction Aware Decoder, GPTDecoder, and DiffusionDecoder. All models are trained and tested on our real‑world robotic dataset. In addition, we introduce a new training set from the RoboTwin 2.0 clean environment (50 tasks, 20 trajectories per task) and a test set from the randomized environment. The training configuration for the decoder remains the same, and the testing setup is consistent with that in our paper.
>
> The experimental results are as follows:
>
> |   Accuracy in Test Set           | Parameters |  RoboTwin  | Real-World Dataset in Paper |
> |---|---|---|---|
> | DINO-Reg + DirectionAwareDecoder | 89.5M      |   **70.72%**   |  **57.13%**  |
> | DINO-Reg + GPTDecoder            | 118.9M     |   48.67%       |  19.43%      |
> | DINO-Reg + DiffusionDecoder      | 90.3M      |   58.78%       |  35.25%      |
>
> Our model outperforms the other two decoders. Together with the results in Section 4.3, these experiments (based on the accuracy comparison under the IDM evaluation) clearly demonstrates the superiority of our Direction Aware Decoder over the commonly used action decoders in VLAs, both in action prediction and embodiment modeling. This outcome further highlights the greater potential of AnyPos in **Embodiment Modeling** via **Task‑Agnostic Exploration** (e.g., acquiring a pre‑trained action decoder from task‑agnostic data that **captures “what is physically feasible and consistent”**), where the action decoder can be used in VLA or unified world models.
>
> [1] Li X, Liu M, Zhang H, et al. Vision-Language Foundation Models as Effective Robot Imitators[C]//The Twelfth International Conference on Learning Representations.

---

> > ### Author Response · Authors · 2025-12-02
> > **Question 6: Analysis of RL, IK, and Rule-Based Data Generation Methods**
> >
> > **Question 6: Analysis of RL, IK, and Rule-Based Data Generation Methods:**
> >
> > **Inverse Kinematics (IK)** offers two common approaches: some IK control algorithms require both end-effector position and rotation to compute joint positions, while others can determine a feasible set of joint positions using only the end-effector position. However, even with the latter approach, the resulting joint positions, while **mathematically valid**, may not always be physically feasible. This infeasibility can arise from factors such as workspace limits, joint constraints, self-collision, environmental collision, and kinematic singularities.
> >
> > In contrast, **reinforcement learning (RL)** policies directly explore feasible joint positions within the whole workspace. By incorporating considerations such as termination conditions and reward functions, RL naturally accounts for workspace boundaries, joint limits, collisions, and singularities. As a result, the action mapping obtained through RL is guaranteed to be feasible, meaning the resulting trajectories are always executable and successful, and they achieve broader workspace coverage.
> >
> > Regarding **rule-based methods**, as mentioned in Appendix Sections A.3 (*Analysis of Exploration Efficiency and Safety*) and A.4 (*Distribution of Task-Agnostic Anypose Dataset and Test Dataset*), we initially implemented a naive trigonometric rule set for task-agnostic random data collection. This simple approach, however, presented several clear limitations:
> > 1. Poor Coverage: By randomly sampling joint positions, many generated values fell outside the range useful for manipulation tasks. Furthermore, the numerous collision-avoidance rules we added overly restricted the action space, resulting in incomplete coverage of feasible actions.
> > 2. Collision: Relying solely on trigonometric solutions made it extremely difficult to prevent collisions given the complex structure of the robotic arms. Despite implementing extensive constraints, the two arms still frequently collided with each other.
> > 3. Off-Screen Motions: Similarly, due to the limitations of the trigonometric rules, the robotic arms would occasionally move outside the camera's field of view.

---

### Official Review · Reviewer_oDT4 · 2025-11-01

**Soundness:** 2
**Presentation:** 1
**Contribution:** 2
**Rating:** 4
**Confidence:** 3

**Summary:**

The paper proposed a novel method to collect random exploration data to learn good representation for the bi-manual manipulation tasks. THe method use a biased sampling stragtegy for exploration and learns a good representation via inverse dynamics model. Then the model is used to learn downstream imitation learning policy. Experiments show better performance than learning from scratch with pure teleoperation data and previous approach.

**Strengths:**

1. The method allows random exploration without direct human supervision, which relax the requirement on human demonstration.

**Weaknesses:**

1. The tasks are not safety critical. The method is hard to generalize to general-purpose tasks, like ones involving safety concerns. Random exploration is not applicable.
2. The writing is poor. The beginning of the paper involves too much distraction of mathematical formulation. It is more straightforward to describe the method in an intuitive way.
3. No ablation on the model design. Why all components are needed?

**Questions:**

1. How's each component contribute to the final performance? How does baselines implemented? Do they similar architecture?

---

> ### Author Response · Authors · 2025-11-23
> **(Updated responses for Weakness 3 and Question 1)**
>
> Thank you for your time and positive feedback! Please find our response below.
>
> ## W1. Safety Concerns and Generalization Abilities
> We appreciate the reviewer's concerns regarding the safety and generalization of our method. At its core, AnyPos learns an embodiment-specific, task-agnostic, and scene-agnostic action mapping in **Task-agnostic Exploration** stage. This design inherently supports generalization to a wide range of tasks, as the mapping is grounded in **embodiment modeling** rather than any specific task or scene. By enabling the collection of task-independent data across diverse scenarios, AnyPos employs random exploration to ensure broad workspace coverage. In practice, the robot performs task-agnostic data collection with either no obstacles or only movable obstacles placed in front of it. This setup ensures that even if collisions occur, they do not cause damage to the robot, any objects, or pose risks to human safety, thereby addressing safety concerns.
>
> ## W2. Writing Improvements
> We sincerely thank the reviewer for their feedback on the writing of this paper. We acknowledge the writing issues and will undertake a thorough revision of the manuscript to significantly enhance its clarity and readability. Regarding the mathematical formulations, we appreciate this insight and will revise these sections to provide more intuitive explanations upfront. The formulas were initially intended to offer a universal and formal description of our motivation and to eliminate ambiguity, but we will strive for a better balance between rigor and accessibility in the final version.
>
> ## W3 & Q1. Ablation Studies and Baseline Implementation
> We have conducted extensive ablation studies to evaluate each component's contribution. The baselines were implemented and trained under the same conditions and data regimes as our method to ensure a fair comparison.
> 1. **Data collection (Sec. 4.2)**: We collected a human-teleoperated dataset comprising 33k image-action pairs from manipulation tasks to serve as a performance benchmark. As a result, our method, trained on the AnyPos dataset, achieves test accuracy comparable to models trained on this human data. In real-world replay testing, our task-agnostic data, owing to its broader coverage of feasible actions, achieves a **33% higher success rate** in replay tasks compared to data collected by humans (Fig. 4b, Fig. 5, Fig. 12).
> 2. **Model Components (Sec. 4.3)**: We performed ablation studies on the arm-decoupling design and the Direction-Aware Decoder (DAD). As detailed in Fig. 4a, our experiments on action prediction accuracy demonstrate that the Arm-Decoupled Estimation module alone improves accuracy by **~20%**, and the DAD module further boosts it by **~21%**. These results underscore the critical importance of each component in our model design.
>
> **Implementation of Baseline Models**: All models consist of a visual encoder paired with an action decoder. The implementation details of both our proposed models and the baseline models are described in Section 4.3, Appendix B.3, and "More Comparisons of Action Decoders" below. This means that different baseline models are constructed by replacing either the visual encoder or the action decoder, while keeping all other configurations unchanged (e.g., hyper-parameters of training).

---

> ### Author Response · Authors · 2025-12-02
> **Weaknesses 3: More Comparisons of Action Decoders**
>
> We sincerely thank the reviewers once again for their insightful questions and constructive feedback. We would like to restate the core abstract idea of our paper: AnyPos consists of two stages: **task-agnostic exploration** and **embodiment modeling**. Task-agnostic exploration is designed to discover “what is physically feasible and consistent,” while embodiment modeling learns all feasible actions, thereby decoupling it from the high-level “how to achieve a goal” question. This decoupling allows both high-level goal-conditioned models and AnyPos to be supervised with ample data in isolation. The IDM (visual encoder + action decoder) serves as one way to validate the effectiveness of embodiment modeling. In the future, we expect that task-agnostic embodiment modeling will help improve the generalization capabilities of models such as VLAs and world models.
> Here, we provide further comparison between our proposed Direction Aware Decoder and two types of action decoders used in VLAs.
>
> **More Comparisons of Action Decoders (II. Embodiment Modeling in Fig.1)**: We further conduct an experiment on the action decoder to evaluate the effectiveness of our Direction Aware Decoder—compared to other decoders—in predicting actions based on visual features, which reflects the quality of embodiment modeling in AnyPos. RoboFlamingo [1] is a well-known vision‑language‑action model composed of a VLM combined with different action decoders (policy heads). We select the **GPTDecoder and DiffusionDecoder from RoboFlamingo as our baselines**. Specifically, we adopt the strongest visual encoder identified in our experiments—DINOv2 with register—and compare three action decoders: the Direction Aware Decoder, GPTDecoder, and DiffusionDecoder. All models are trained and tested on our real‑world robotic dataset. In addition, we introduce a new training set from the RoboTwin 2.0 clean environment (50 tasks, 20 trajectories per task) and a test set from the randomized environment. The training configuration for the decoder remains the same, and the testing setup is consistent with that in our paper.
>
> The experimental results are as follows:
>
> |   Accuracy in Test Set           | Parameters |  RoboTwin  | Real-World Dataset in Paper |
> |---|---|---|---|
> | DINO-Reg + DirectionAwareDecoder | 89.5M      |   **70.72%**   |  **57.13%**  |
> | DINO-Reg + GPTDecoder            | 118.9M     |   48.67%       |  19.43%      |
> | DINO-Reg + DiffusionDecoder      | 90.3M      |   58.78%       |  35.25%      |
>
> Our model outperforms the other two decoders. Together with the results in Section 4.3, these experiments (based on the accuracy comparison under the IDM evaluation) clearly demonstrates the superiority of our Direction Aware Decoder over the commonly used action decoders in VLAs, both in action prediction and embodiment modeling. This outcome further highlights the greater potential of AnyPos in **Embodiment Modeling** via **Task‑Agnostic Exploration** (e.g., acquiring a pre‑trained action decoder from task‑agnostic data that **captures “what is physically feasible and consistent”**), where the action decoder can be used in VLA or unified world models.
>
> [1] Li X, Liu M, Zhang H, et al. Vision-Language Foundation Models as Effective Robot Imitators[C]//The Twelfth International Conference on Learning Representations.

---

### Official Review · Reviewer_dmk8 · 2025-11-01

**Soundness:** 3
**Presentation:** 2
**Contribution:** 3
**Rating:** 6
**Confidence:** 3

**Summary:**

The paper introduces an embodiment modeling framework that learns a vision-to-action mapping directly from large-scale, automatically collected trajectories. It employs an RL-based explorer to sample EEF targets across the 3D workspace and convert them into feasible joint configurations, creating a large dataset. The inverse dynamics model trained on this data maps visual observations to joint positions with good precision, aided by an arm-decoupled architecture and a Direction-Aware Decoder. The system can then pair this embodiment model with high-level policies such as video-generation or VLA models, achieving real-world bimanual manipulation success while requiring no human teleoperation.

**Strengths:**

The paper provides a large scale dataset by RL-based exploration, and then recording 610k safe, diverse image-action pairs in 10 hours, which is claimed to be 30× faster than human teleoperation.

The quantitative results for real-world validation looks strong, with high replay success, the demo video also shows performance, though with some failure cases.

The paper presents detailed implementation and hyperparameter reporting.

**Weaknesses:**

The paper does not provide comparison when replacing learned inverse dynamics model (IDM) with traditional collision-free inverse kinematics (IK). most baselines are vision encoders (ResNet or DINOv2) rather than control pipelines, so the benchmark scope is narrow. Without such an ablation, it is unclear whether the learned model actually outperforms or simply replicates IK performance under noise and multi-arm constraints, since IK is guaranteed to be generalizable but a learned model is not

Variables are introduced abruptly in introduction without clear linkage to the system. The overall abstraction and introduction could better define these terms and explicitly describe the model’s input and output, integrating this explanation with Figure 1 to clarify the pipeline flow.

The cross embodiment experiment is limited given the paper claims embodiment-agnostic modeling

Arms overlap with objects or arms or move out of frame -- conditions that are common in cluttered manipulation scenarios, which the model seems not be able to address.

**Questions:**

It seems AnyPos depends only on URDF/kinematics and can be “replayed” for new viewpoints. How robust is the IDM to camera shifts?

---

> ### Author Response · Authors · 2025-11-23
>
> Thank you for your time and positive feedback! Please find our response below.
>
> ## 1. Lack of Comparison with IK
> We would like to clarify that the learned IDM and traditional IK address fundamentally distinct challenges. Traditional IK solves for joint positions given a target end-effector pose, operating purely in the kinematic domain. In contrast, our IDM is designed to translate **visual observations** directly into feasible joint actions. Given that the two methods operate on different inputs and solve different sub-problems, a direct comparison would be asymmetric.
>
> ## 2. Clarification on Terminology and Pipeline Flow
> Thank you for this constructive feedback.
> 1. **Writing issues and revision plans**: We acknowledge the writing issues and will undertake a thorough revision of the manuscript to significantly enhance its clarity and readability. In the revised manuscript, we plan to: (1) provide clearer definitions of all key terms at their first appearance; (2) explicitly list the model's inputs and outputs in the introduction; and (3) thoroughly revise the text to integrate the explanation with Figure 1, ensuring the figure is utilized to best illustrate the flow of our pipeline.
> 2. **Clarification on pipeline flow**: Our pipeline involves two steps: (1) obtain a task-agnostic training dataset covering the entire cubic workspace of dual robotic arms through our automated data collection method, and (2) train an embodiment model using this dataset.
>
> ## 3. Clarification on Embodiment-Agnostic Experiment
> We appreciate the reviewer's comment and would like to clarify that **our work does not claim AnyPos to be embodiment-agnostic**; instead, in this paper, we propose **task-agnostic embodiment modeling** as a solution to address robot manipulation data scarcity and mitigate cross-embodiment challenges, with AnyPos serving as a unified pipeline that combines large-scale automated data collection with robust model learning.
> While this approach is embodiment-specific, it can be trained from scratch, including both data collection and model training, in a few hours (e.g., 5~10 hours + 1 day for our setup), demonstrating its practicality in handling different embodiment requirements.
> Furthermore, we demonstrate that AnyPos can be straightforwardly adapted to a new embodiment, as validated on RoboTwin (App. A.1, Tab. 3), which employs a different morphology and environment. This adaptation requires data collection and training but is achieved with minimal effort. Quantitatively, a VGM+IDM pipeline using AnyPos as the IDM outperformed SOTA methods like RDT and Pi0 by 32% and 23%, respectively, on 17 tasks in the clean setting.
>
> ## 4. Addressing Arm Visibility in Cluttered Scenarios
> We acknowledge that scenarios involving significant arm-object occlusion or the arm moving outside the camera frame represent a current limitation of our model. To mitigate this issue, AnyPos resets the intermediate camera to a rear elevated position (Sec. 4.1) to capture a complete view of the robotic arm. After this camera reset, the arm remains within the frame while performing tabletop tasks. Despite its effectiveness in our current tasks, we recognize that certain edge cases may still arise where the arm moves out of view, leading to embodiment modeling failures. Therefore, it remains an objective for our future work to address such scenarios, improve the model's robustness, and enhance its applicability to more complex real-world tasks.
> We have conducted preliminary validation showing that the IDM can be effectively learned even when the robotic arm is out of view or occluded, **as long as task-aware data is used** for training. However, the performance of embodiment modeling under the same visual conditions but **with task-agnostic data** remains unverified. We intend to systematically assess this aspect using the RobotWin benchmark. We will update the results once the experiments are completed.
>
> ## 5. Robustness to Camera Viewpoint Changes
> AnyPos is robust as long as the camera remains stationary relative to its original fixed position. This is because the visual observations and their relationship to the robot's proprioceptive state remain coherent. However, retraining is required for unseen robot morphologies, as our embodiment model is embodiment-specific. Despite the need for retraining, it is a quick process. A full retraining cycle from scratch, including automated data collection and model training, takes just 5~10 hours for data collection and approximately a day for model training as mentioned before.

---

> ### Author Response · Authors · 2025-12-02
> **Weakness 4: Arm Occlusion in Cluttered Environments**
>
> **Arm Occlusion in Cluttered Environments**: In the original setup used in the paper, we employed a rear-mounted elevated camera to maintain a complete view of the robotic arm. To address your question—specifically, to demonstrate that our IDM can still perceive the situation and successfully complete the task even when the robotic arm overlaps with objects or moves out of the frame—we revert to the default front-mounted camera configuration at the standard height (as in the official RoboTwin setup). In this setting, the robotic arm frequently leaves the frame or becomes occluded by objects. We evaluate task success rates on RoboTwin tasks under these conditions, keeping all other settings unchanged.
> The experimental results are summarized in the table below:
>
> | Task / Success Rate (%)   | AnyPos(Ours) | Anypos_occ | RDT   | Pi0   | ACT   | DP    | DP3     |
> | ------------------------- | ------------ | ---------- | ----- | ----- | ----- | ----- | ------- |
> | Adjust Bottle             | 95           | 70         | 81    | 90    | 97    | 97    | **99**  |
> | Click Alarmclock          | **100**      | 100        | 61    | 63    | 32    | 61    | 77      |
> | Click Bell                | 95           | **100**    | 80    | 44    | 58    | 54    | 90      |
> | Grab Roller               | **100**      | 95         | 74    | 96    | 94    | 98    | 98      |
> | Lift Pot                  | 75           | **100**    | 72    | 84    | 88    | 39    | **97**  |
> | Move Can Pot              | 50           | **75**     | 25    | 58    | 22    | 39    | **70**  |
> | Move Pillbottle Pad       | 70           | **80**     | 8     | 21    | 0     | 1     | 41      |
> | Move Playingcard Away     | **100**      | 95         | 43    | 53    | 36    | 47    | 68      |
> | Pick Dual Bottles         | 75           | **80**     | 42    | 57    | 31    | 24    | 60      |
> | Place Container Plate     | **100**      | 95         | 78    | 88    | 72    | 41    | 86      |
> | Place Empty Cup           | **100**      | 85         | 56    | 37    | 61    | 37    | 65      |
> | Place Object Stand        | **95**       | 80         | 15    | 36    | 1     | 22    | 60      |
> | Press Stapler             | 90           | **100**    | 41    | 62    | 31    | 6     | 69      |
> | Shake Bottle              | **100**      | 100        | 74    | 97    | 74    | 65    | 98      |
> | Shake Bottle two          | 85           | **100**    | 76    | 91    | 82    | 61    | 83      |
> | Shake Bottle Horizontally | **100**      | **100**    | 84    | 99    | 63    | 59    | **100** |
> | Turn Switch               | **70**       | 50         | 35    | 27    | 5     | 36    | 46      |
> | **Average Success Rate**  | **88.24**    | **88.53**      | 55.59 | 64.88 | 49.82 | 46.29 | 76.88   |
>
> Here, AnyPos (Ours) refers to the setup with the rear-mounted elevated camera, while **AnyPos_occ** denotes the setup with the default front-mounted camera at the standard height—where the arm is subject to occlusion or frequently moves out of view. As shown in the table, even when the camera is adjusted to a position that introduces occlusion, our pipeline achieves task success rates that are comparable to those in the original setup. This is because during the key execution steps of each task, the robotic arm inevitably appears in the field of view of the middle camera. As long as these critical frames can be correctly decoded into actions by the IDM, the task can still be completed successfully.

---

> > ### Author Response · Authors · 2025-12-02
> > **More Comparisons of Action Decoders (II. Embodiment Modeling in Fig.1)**
> >
> > We sincerely thank the reviewers once again for their insightful questions and constructive feedback. We would like to restate the core abstract idea of our paper: AnyPos consists of two stages: **task-agnostic exploration** and **embodiment modeling**. Task-agnostic exploration is designed to discover “what is physically feasible and consistent,” while embodiment modeling learns all feasible actions, thereby decoupling it from the high-level “how to achieve a goal” question. This decoupling allows both high-level goal-conditioned models and AnyPos to be supervised with ample data in isolation. The IDM (visual encoder + action decoder) serves as one way to validate the effectiveness of embodiment modeling. In the future, we expect that task-agnostic embodiment modeling will help improve the generalization capabilities of models such as VLAs and world models.
> > Here, we provide further comparison between our proposed Direction Aware Decoder and two types of action decoders used in VLAs.
> >
> > **More Comparisons of Action Decoders (II. Embodiment Modeling in Fig.1)**: We further conduct an experiment on the action decoder to evaluate the effectiveness of our Direction Aware Decoder—compared to other decoders—in predicting actions based on visual features, which reflects the quality of embodiment modeling in AnyPos. RoboFlamingo [1] is a well-known vision‑language‑action model composed of a VLM combined with different action decoders (policy heads). We select the **GPTDecoder and DiffusionDecoder from RoboFlamingo as our baselines**. Specifically, we adopt the strongest visual encoder identified in our experiments—DINOv2 with register—and compare three action decoders: the Direction Aware Decoder, GPTDecoder, and DiffusionDecoder. All models are trained and tested on our real‑world robotic dataset. In addition, we introduce a new training set from the RoboTwin 2.0 clean environment (50 tasks, 20 trajectories per task) and a test set from the randomized environment. The training configuration for the decoder remains the same, and the testing setup is consistent with that in our paper.
> >
> > The experimental results are as follows:
> >
> > |   Accuracy in Test Set           | Parameters |  RoboTwin  | Real-World Dataset in Paper |
> > |---|---|---|---|
> > | DINO-Reg + DirectionAwareDecoder | 89.5M      |   **70.72%**   |  **57.13%**  |
> > | DINO-Reg + GPTDecoder            | 118.9M     |   48.67%       |  19.43%      |
> > | DINO-Reg + DiffusionDecoder      | 90.3M      |   58.78%       |  35.25%      |
> >
> > Our model outperforms the other two decoders. Together with the results in Section 4.3, these experiments (based on the accuracy comparison under the IDM evaluation) clearly demonstrates the superiority of our Direction Aware Decoder over the commonly used action decoders in VLAs, both in action prediction and embodiment modeling. This outcome further highlights the greater potential of AnyPos in **Embodiment Modeling** via **Task‑Agnostic Exploration** (e.g., acquiring a pre‑trained action decoder from task‑agnostic data that **captures “what is physically feasible and consistent”**), where the action decoder can be used in VLA or unified world models.
> >
> > [1] Li X, Liu M, Zhang H, et al. Vision-Language Foundation Models as Effective Robot Imitators[C]//The Twelfth International Conference on Learning Representations.

---

### Official Review · Reviewer_EEKr · 2025-11-01

**Soundness:** 3
**Presentation:** 3
**Contribution:** 3
**Rating:** 6
**Confidence:** 4

**Summary:**

This paper introduces AnyPos, a framework for task-agnostic embodiment modeling in robotic manipulation. It addresses the critical challenges of data scarcity, task-specificity, and poor cross-platform generalization in robot learning. The core innovation lies in decoupling the learning of physically feasible actions from high-level task semantics. The paper achieve this through a two-stage pipeline: First, an automated, safety-aware exploration process collects a dataset of diverse, feasible robot trajectories without human teleoperation. Second, an inverse dynamics model is trained on this data, employing arm-decoupled estimation and a direction-aware decoder to achieve high-precision action prediction robust to distribution shifts. The resulting embodiment model serves as a reusable "motion prior" that can be seamlessly coupled with various high-level policy models. Experiments show a good improvement in action prediction accuracy and real-world task success rates over strong baselines.

**Strengths:**

S1) The paper is well written, the proposed pipeline is simple and easy to follow.

S2) The automated data collection framework demonstrates a highly effective strategy for generating large-scale robotic datasets.

S3) The design of arm-decoupled estimation and the direction-aware decoder directly addresses the challenges of high-dimensional action spaces and visual ambiguity in task-agnostic data. These components are empirically shown to be critical for achieving the high-precision action prediction required for real-world deployment.

**Weaknesses:**

W1) I concerns the practical deployment and generalization of the automated data collection framework. My primary question revolves around the necessity and process of training the RL-based projection policy (`f_RL`). To clarify its scope and limitations: when encountering a new physical scene with different objects and layouts, must a new RL policy be trained from scratch in a simulation that explicitly reconstructs that specific bounded workspace volume? Furthermore, was a single, universal RL policy used to collect all the task-agnostic data for the diverse tasks in the paper, or were multiple scene-specific policies required? Ultimately, I seek to understand the inherent limitations of this approach, specifically, how a policy trained on one scene is expected to perform when deployed in a novel, unseen environment without retraining, particularly regarding its ability to avoid collisions and maintain feasibility.

W2) While the paper's core contribution lies in using improved data to train a more robust inverse dynamics model, the proposed "task-gnostic embodiment modeling" framework ultimately relies on a pipeline combining a video generation model (VGM) with this learned model. This architecture appears to have limited practical utility due to its inherent susceptibility to error propagation. The system's performance is heavily contingent on the quality of the generated videos, where any inaccuracies in the predicted future frames are directly translated into action errors by the inverse dynamics model. A shortcoming of the work is the lack of a detailed analysis quantifying this compounding error and visualizing the pipeline's robustness.

**Questions:**

Please see weaknesses.

---

> ### Author Response · Authors · 2025-11-23
>
> Thank you for your time and positive feedback! Please find our response below.
>
> ## 1. Task-Agnostic Embodiment-aware Data Collection Framework
> We acknowledge the reviewer's concern regarding the real-world application and adaptability of our automated data collection system. To clarify:
>
> 1. For a new physical scene with different objects and layouts, **no retraining is needed**.
> 2. A **single, universal RL policy** was used to collect all the task-agnostic data for the diverse tasks in the paper.
>     - This policy is embodiment-specific, task-agnostic, and scene-agnostic.
>
> To further clarify its scope and limitations:
>
> - **Collection Workflow Clarification**: Our RL-based task-agnostic embodiment-aware data collection framework works as follows: (0) for any new robot embodiment; (1) first, we train a RL policy using the URDF of the specific embodiment in the simulator; (2) second, we leverage the trained RL policy to construct a mapping from end-effector positions to feasible joint positions within a bounded workspace volume; (3) third, we use the embodiment-specific mapping to direct a uniform exploration of the workspace; (4) fourth, we construct a task-agnostic dataset by collecting pairs of observations and corresponding actions during the exploration.
>     - Note that *the bounded workspace volume mentioned in (2) refers to the reachable workspace of the robotic manipulator*, which is dependent on the robot embodiment but independent of specific scenes or tasks.
>     - Note that *the mapping mentioned in (2) is embodiment-specific, yet task-agnostic and scenario-agnostic*, which enables the collection of task-independent data for that embodiment across any scenario.
> - **Cross-Environment Generalization without Retraining**: Given that the RL policy is only used for mapping end-effector positions to feasible joint positions, and that **the policy is embodiment-specific, yet task-agnostic and scenario-agnostic**, no retraining is required for a new physical scene with different objects and layouts.
> - **Safety and Feasibility**: Given that the primary objective of our data collection framework is to collect data for embodiment modeling, the focus during data collection is to achieve uniform coverage of the entire workspace. In practice, the robot performs random exploration with either no obstacles or only movable obstacles placed in front of it. This setup ensures that even if collisions occur, they do not cause damage to the robot, any objects, or pose risks to human safety. By addressing safety concerns in this manner, we are able to efficiently gather data that comprehensively covers the workspace.
>
> ## 2. Method Focus and Empirical Validation
> We thank the reviewer for the opportunity to clarify the key aspects of our method's core contribution and model performance.
> 1. **Core Contribution**: We would like to clarify that our core contribution is the proposed task-agnostic embodiment modeling approach (AnyPos), rather than the VGM+IDM pipeline itself, which we used to empirically demonstrate a potential application of AnyPos as a robust inverse dynamics model and to validate its embodiment modeling abilities.
> 2. **Model Performance**: We acknowledge the reviewer's valid concern regarding potential error propagation in the VGM+IDM pipeline. To directly address the concern regarding error propagation, we are planning an ablation study that replaces the generated videos from the VGM with ground-truth videos of the same tasks (with the same reset). Specifically, by comparing the task success rates achieved using ground-truth videos versus generated videos as input to AnyPos, we will be able to precisely isolate and quantify the compounding error introduced by the video generation step. We will update the results once the experiments are completed.

---

> ### Author Response · Authors · 2025-12-02
> **More Comparisons of Action Decoders in VLAs**
>
> We sincerely thank the reviewers once again for their insightful questions and constructive feedback. We would like to restate the core abstract idea of our paper: AnyPos consists of two stages: **task-agnostic exploration** and **embodiment modeling**. Task-agnostic exploration is designed to discover “what is physically feasible and consistent,” while embodiment modeling learns all feasible actions, thereby decoupling it from the high-level “how to achieve a goal” question. This decoupling allows both high-level goal-conditioned models and AnyPos to be supervised with ample data in isolation. The IDM (visual encoder + action decoder) serves as one way to validate **the effectiveness of embodiment modeling**. In the future, we expect that task-agnostic embodiment modeling will help improve the generalization capabilities of models such as VLAs and world models.
> Here, we provide further comparison between our proposed Direction Aware Decoder and two types of action decoders used in VLAs.
>
> **More Comparisons of Action Decoders (II. Embodiment Modeling in Fig.1)**: We further conduct an experiment on the action decoder to evaluate the effectiveness of our Direction Aware Decoder—compared to other decoders—in predicting actions based on visual features, which reflects the quality of embodiment modeling in AnyPos. RoboFlamingo [1] is a well-known vision‑language‑action model composed of a VLM combined with different action decoders (policy heads). We select the **GPTDecoder and DiffusionDecoder from RoboFlamingo as our baselines**. Specifically, we adopt the strongest visual encoder identified in our experiments—DINOv2 with register—and compare three action decoders: the Direction Aware Decoder, GPTDecoder, and DiffusionDecoder. All models are trained and tested on our real‑world robotic dataset. In addition, we introduce a new training set from the RoboTwin 2.0 clean environment (50 tasks, 20 trajectories per task) and a test set from the randomized environment. The training configuration for the decoder remains the same, and the testing setup is consistent with that in our paper.
>
> The experimental results are as follows:
>
> |                                  | Parameters |  RoboTwin  | Real-World Dataset in Paper |
> |---|---|---|---|
> | DINO-Reg + DirectionAwareDecoder | 89.5M      |   **70.72%**   |  **57.13%**  |
> | DINO-Reg + GPTDecoder            | 118.9M     |   48.67%       |  19.43%      |
> | DINO-Reg + DiffusionDecoder      | 90.3M      |   58.78%       |  35.25%      |
>
> Our model outperforms the other two decoders. Together with the results in Section 4.3, these experiments (based on the accuracy comparison under the IDM evaluation) clearly demonstrates the superiority of our Direction Aware Decoder over the commonly used action decoders in VLAs, both in action prediction and embodiment modeling. This outcome further highlights the greater potential of AnyPos in **Embodiment Modeling** via **Task‑Agnostic Exploration** (e.g., acquiring a pre‑trained action decoder from task‑agnostic data that **captures “what is physically feasible and consistent”**), where the action decoder can be used in VLA or unified world models.
>
> [1] Li X, Liu M, Zhang H, et al. Vision-Language Foundation Models as Effective Robot Imitators[C]//The Twelfth International Conference on Learning Representations.

---

> ### Author Response · Authors · 2025-12-02
> **More experiments about weakness 2: ground truth video+AnyPos vs. VGM+AnyPos**
>
> **Quantifying the Compounding Error of VGM+AnyPos**: To quantitatively analyze the compounding error in our proposed VGM+AnyPos framework and verify that our Inverse Dynamics Model (IDM) can accurately predict actions not only from ground truth videos but also from generated videos, we conduct a comparative evaluation between the ground truth video+AnyPos and VGM+AnyPos pipelines.
> In the RoboTwin simulator, both methods are evaluated under identical environment settings. We first use the official data collection strategy to obtain ground truth videos, then reset the environment to its initial state to measure the task success rates of the two approaches.
>
> | Task / Success Rate (%)   | AnyPos(Ours) | Anypos_gt | RDT   | Pi0    | ACT   | DP    | DP3     |
> | ------------------------- | ------------ | --------- | ----- | ------ | ----- | ----- | ------- |
> | Adjust Bottle             | 95           | **100**   | 81    | 90     | 97    | 97    | **99**  |
> | Click Alarmclock          | **100**      | **100**   | 61    | 63     | 32    | 61    | 77      |
> | Click Bell                | 95           | **100**   | 80    | 44     | 58    | 54    | 90      |
> | Grab Roller               | **100**      | **100**   | 74    | 96     | 94    | 98    | 98      |
> | Lift Pot                  | 75           | **100**   | 72    | 84     | 88    | 39    | 97      |
> | Move Can Pot              | 50           | **90**    | 25    | 58     | 22    | 39    | 70      |
> | Move Pillbottle Pad       | 70           | **100**   | 8     | 21     | 0     | 1     | 41      |
> | Move Playingcard Away     | **100**      | **100**   | 43    | 53     | 36    | 47    | 68      |
> | Pick Dual Bottles         | 75           | **100**   | 42    | 57     | 31    | 24    | 60      |
> | Place Container Plate     | **100**      | 95        | 78    | 88     | 72    | 41    | 86      |
> | Place Empty Cup           | **100**      | **100**   | 56    | 37     | 61    | 37    | 65      |
> | Place Object Stand        | 95           | **100**   | 15    | 36     | 1     | 22    | 60      |
> | Press Stapler             | 90           | **100**   | 41    | 62     | 31    | 6     | 69      |
> | Shake Bottle              | **100**      | **100**   | 74    | 97     | 74    | 65    | 98      |
> | Shake Bottle two          | 85           | **90**    | 76    | **91** | 82    | 61    | 83      |
> | Shake Bottle Horizontally | **100**      | 95        | 84    | 99     | 63    | 59    | **100** |
> | Turn Switch               | 70           | **80**    | 35    | 27     | 5     | 36    | 46      |
> | **Average Success Rate**  | 88.24        | **97.06** | 55.59 | 64.88  | 49.82 | 46.29 | 76.88   |
>
> In this context, "AnyPos (Ours)" refers to the full VGM+AnyPos pipeline, while "AnyPos_gt" uses ground truth video for inference with the AnyPos framework. As shown, the success rate for ground truth video+AnyPos is slightly higher than that for VGM+AnyPos. Notably, the ground truth video+AnyPos approach achieves​ a success rate of **nearly 100%** (97.06%). This aligns with the real-world replay experiments reported in our paper: when the input video is sufficiently accurate, the IDM's predicted actions lead to near-perfect task execution, indicating that the **IDM's own error is negligible**. Therefore, the compounding error primarily originates from the VGM. Furthermore, both pipelines significantly outperform all other baselines, achieving success rates above 88%. This demonstrates the **strong robustness of our IDM** in accurately interpreting generated videos.
>
> Additionally, we reiterate that our AnyPos framework consists of two distinct stages: *task-agnostic exploration* and *embodiment modeling*. Both stages are fundamentally concerned with learning *"what is physically feasible and consistent."* The IDM serves as one method to validate the effectiveness of the embodiment modeling stage. We anticipate that task-agnostic embodiment modeling will help address generalization challenges in future work on Video-Language-Action models (VLAs) and world models. And our preliminary experiment titled "More Comparisons of Action Decoders in VLAs," which compares our proposed Direction-Aware Decoder against two common action decoders in VLAs, offers an important insight: **the action decoder in a VLA is essentially an embodiment model**. This suggests that effective *embodiment modeling can be learned from task-agnostic data*.

---

### Meta-Review · Area_Chair_hyCB · 2026-01-07

**Summary:**

This paper focuses on the problem of task-agnostic embodied modeling. The authors argue that by decoupling task constraints and expanding coverage of the state–action space, manipulation policies can reduce dependence on narrow goal annotations and transfer more effectively across tasks, embodiments, and viewpoints. To achieve this, the paper proposes a two-stage pipeline featuring automated task-agnostic exploration for data collection and an inverse dynamics model trained on the collected data. Experiments conducted in both simulation and the real world demonstrate improvements compared to several baselines.

The paper received mixed scores, with two positive and two negative ratings. Most reviewers acknowledged the strong motivation for task-agnostic learning and the design of the two-stage pipeline. However, Reviewers EEKr, oDT4, and kAye raised concerns regarding the RL components and safety issues involved in automated task-agnostic exploration. Additionally, Reviewer EEKr questioned the scalability of task-agnostic learning, while Reviewer kAye argued that the experiments did not sufficiently demonstrate the cross-task, cross-view, and cross-embodiment capabilities of AnyPos. Reviewer dmk8 and oDT4 suggested that the writing requires improvement.

The rebuttal addressed most of the concerns; however, the meta-reviewer noted that two main issues remain unresolved.
1. Framework scalability of task-agnostic learning (Reviewer EEKr W2). There are no experiments demonstrating that AnyPos is effective when applied to frameworks other than the VGM + inverse dynamics model.
2. Insufficient experiments (Reviewer kAye). The authors did not provide extensive experimental results or theoretical analysis to validate the true benefits of task-agnostic embodied modeling.
Overall, the meta-reviewer recommends rejecting this paper and suggests that the authors reorganize their contributions and provide further validation of the effectiveness of task-agnostic embodied modeling.

**Reviewer Concerns:**

Addressed Concerns:
1. Practicality of automated data collection (Reviewer EEKr, kAye). The authors addressed this concern by clarifying the operational workflow and providing comprehensive details regarding the random exploration strategy.
2. Comparison with Inverse Kinematics (Reviewer dmk8). The authors clarified that Inverse Kinematics (IK) and the inverse dynamics model address fundamentally different challenges.
3. Ablation study (Reviewer oDT4). The authors addressed this by highlighting the ablation studies performed on both data collection and model components within the paper.

Outstanding Concerns:
1. Framework scalability of task-agnostic learning (Reviewer EEKr). There is insufficient evidence to demonstrate that AnyPos can effectively transfer to other frameworks.
2. Insufficient experiments (Reviewer kAye). Lack of experiments or analysis demonstrating the method's capability in cross-view, cross-task, or cross-embodiment settings.

**Reviewer Scores:**

Based on the rebuttal and discussion phase, I anticipate that
1. Reviewers dmk8, EEKr would likely maintain their original scores 6.
2. Reviewer oDT4 would likely raise the score to 6 because the concerns have been addressed.
3. Reviewer kAye will maintain the score of 4 as the primary concern remain partially unresolved.

---

### Decision · Program_Chairs · 2026-01-26

Reject